# The evolutionary plasticity of chromosome metabolism allows adaptation to constitutive DNA replication stress

Marco Fumasoni*, Andrew W Murray

Department of Molecular and Cellular Biology, Harvard University, Cambridge, United States

**Abstract** Many biological features are conserved and thus considered to be resistant to evolutionary change. While rapid genetic adaptation following the removal of conserved genes has been observed, we often lack a mechanistic understanding of how adaptation happens. We used the budding yeast, *Saccharomyces cerevisiae*, to investigate the evolutionary plasticity of chromosome metabolism, a network of evolutionary conserved modules. We experimentally evolved cells constitutively experiencing DNA replication stress caused by the absence of Ctf4, a protein that coordinates the enzymatic activities at replication forks. Parallel populations adapted to replication stress, over 1000 generations, by acquiring multiple, concerted mutations. These mutations altered conserved features of two chromosome metabolism modules, DNA replication and sister chromatid cohesion, and inactivated a third, the DNA damage checkpoint. The selected mutations define a functionally reproducible evolutionary trajectory. We suggest that the evolutionary plasticity of chromosome metabolism has implications for genome evolution in natural populations and cancer.

*For correspondence:
marcofumasoni@fas.harvard.edu

Competing interests: The authors declare that no competing interests exist.

## Introduction

The central features of many fundamental biological processes have been conserved since the last common ancestor of all extant organisms. Many of the proteins involved in these processes are essential, and the complex molecular interactions between them have been argued to constrain the evolution of both the processes and the proteins that carry them out (*Hirsh and Fraser, 2001*; *Jordan et al., 2002*; *Wilson et al., 1977*). The strength of these constraints has been questioned by studies that demonstrated that organisms can evolutionary adapt to the removal of important, and sometimes essential cellular genes (*Liu et al., 2015*; *Rancati et al., 2008*). Although the mutations that cause some adaptations have been identified, we lack a mechanistic understanding of how they repair the initial defect. Furthermore, in systematic studies, defects in some processes, such as intracellular trafficking, were more easily repaired, by single genetic events, than others, such as ribosome biogenesis, mRNA synthesis and DNA replication (*Liu et al., 2015*; *van Leeuwen et al., 2016*).

Replication requires multiple enzymes that catalyze individual reactions such as unwinding the double helix, priming replication, and synthesizing new DNA strands (*O'Donnell et al., 2013*). A common feature of replication is the organization of these enzymatic activities in multi-molecular complexes called replisomes, whose function is to coordinate the simultaneous synthesis of DNA from the two anti-parallel template strands (*Yao and O'Donnell, 2016*).

The temporal and physical interactions amongst the enzymatic machines that performs the different steps of DNA replication are remarkably conserved. Nevertheless, differences in many features

**eLife digest** All plants, animals and fungi share a common ancestor, and though they have evolved to become very distinct over billions of years, they all share the essential machinery needed for cells to grow and divide. At the heart of this is the complex interaction of proteins involved in DNA replication, the process of duplicating the genetic material every time a cell divides. DNA replication needs to be done with great care, with error rates as small as one mistake in a billion. Otherwise, mutations can accumulate in the genome, causing problems for long-term survival. Despite the overall principles of DNA replication remaining the same, the underlying mechanisms vary across different organisms. Given the precision and complexity of replicating DNA, it was not clear how the process had evolved mechanistic differences.

Fumasoni and Murray set out to answer this by forcing a strain of budding yeast to evolve by removing the gene for an important, but not essential, component of DNA replication. The cells were still able to reproduce, but they were hampered by this mutation.

Fumasoni and Murray studied the yeast after it had reproduced for a thousand generations, giving it enough time to acquire new mutations that would allow it to compensate for the initial defects. In eight separate samples, the yeast had made many of the same changes in order to overcome the original mutation. These mutations altered conserved features of DNA replication and the segregation of genetic material and inactivated a third feature that would normally protect the cell against the accumulation of damaged DNA.

These findings show how reproducible the evolutionary pathways can be in a controlled, laboratory environment and that cells can evolve quickly after conserved processes in the cell are damaged. The behavior of the mutated yeast mimicked that of cancer cells, which are often struggling to adapt to mutations in their replication machinery. Studying the rapid evolution that follows genetic perturbations could help researchers to better deal with challenges in cancer treatment and the development of antibiotic resistance in bacteria, as well as leading to a deeper understanding of both evolution and cell biology.

of DNA replication have been reported: the number of replisome subunits is higher in eukaryotes than in bacteria, possibly to account for the higher complexity of eukaryotic genomes (*McGeoch and Bell, 2008*). Some subunits are only found in some eukaryotic species (*Aves et al., 2012*; *Liu et al., 2009*). Notably, there are also biochemical variations in important features, such as the helicase, which encircles the leading strand in eukaryotes and the lagging strand in prokaryotes (*McGeoch and Bell, 2008*), or differences in the regulation of DNA replication by the machinery that drives the cell cycle progression (*Cross et al., 2011*; *Parker et al., 2017*; *Siddiqui et al., 2013*).

These differences reveal that although the DNA replication module performs biochemically conserved reactions, its features can change during evolution. This observation poses an apparent paradox: how can such an important process change during evolution without killing cells? One hypothesis is that the overall organization of DNA replication can change as a consequence of accumulating several mutations, each perturbing a single aspect of replication, in response to a severe initial perturbation.

To test this hypothesis, we followed the evolutionary response to a genetic perturbation of DNA replication. Characterizing evolutionary responses to genetic perturbations has informed studies of functional modules (*Filteau et al., 2015*; *Harcombe et al., 2009*; *Rojas Echenique et al., 2019*), challenged the notion that particular genes are essential (*Liu et al., 2015*; *Rancati et al., 2018*), and revealed that initial genotypes can determine evolutionary trajectories (*Lind et al., 2015*; *Rojas Echenique et al., 2019*; *Szamecz et al., 2014*).

We followed the evolutionary response of *S. cerevisiae* to DNA replication stress, an overall perturbation of DNA replication that interferes with chromosome metabolism, reduces cell viability, and induces genetic instability (*Muñoz and Méndez, 2017*; *Zeman and Cimprich, 2014*). DNA replication stress has been implicated in both cancer progression and aging (*Burhans and Weinberger, 2007*; *Gaillard et al., 2015*) but despite studies investigating the direct effect of replication stress on cell physiology, its evolutionary consequences are unknown.

We imposed constitutive replication stress by removing Ctf4, a component of the replisome and evolved eight populations for 1000 generations. We exploited the ability of experimental evolution to identify, analyze, and compare the mutations that create parallel evolutionary trajectories to increase fitness (*Barrick and Lenski, 2013*; *Van den Bergh et al., 2018*). We found that populations can recover from the fitness defect induced by DNA replication stress. Genetic analysis revealed that their adaptation is driven by mutations that damage, alter, and improve conserved features of three modules involved in chromosome metabolism: DNA replication, the DNA damage checkpoint, and sister chromatid cohesion. These mutations arise sequentially and collectively allow cells to approach the fitness of their wild-type ancestors within 1000 generations of evolution. The molecular basis of these adaptive strategies and their epistatic interactions produce a mechanistic model of the evolutionary adaptation to replication stress. Our results reveal the short-term evolutionary plasticity of chromosome metabolism. We discuss the consequences of this plasticity for the evolution of species in the wild and cancer progression.

## Results

### Adaptation to DNA replication stress is driven by mutations in chromosome metabolism

Replication stress refers to the combination of the defects in DNA metabolism and the cellular response to these defects in cells whose replication has been substantially perturbed (*Macheret and Halazonetis, 2015*). Problems in replication can arise at the sites of naturally occurring or experimentally induced lesions and can cause genetic instability (*Muñoz and Méndez, 2017*). We asked how cells evolve to adapt to constitutive DNA replication stress.

Previous work has induced replication stress by using chemical treatments or genetic perturbations affecting factors involved in DNA replication (*Mazouzi et al., 2016*; *Tkach et al., 2012*; *Zheng et al., 2016*). To avoid evolving resistance to drugs or the reversion of point mutations that induce replication stress, we chose instead to remove *CTF4*, a gene encoding an important, but non-essential, component of the DNA replication machinery. Ctf4 is a homo-trimer, that serves as a structural hub within the replisome and coordinates different aspects of DNA replication by binding the replicative helicase, the primase, and other factors recruited to the replication fork (*Figure 1A*; *Gambus et al., 2009*; *Samora et al., 2016*; *Simon et al., 2014*; *Tanaka et al., 2009*; *Yuan et al., 2019*). In the absence of Ctf4, cells experience several problems in fork progression leading to the accumulation of defects commonly associated with DNA replication stress (*Muñoz and Méndez, 2017*), such as single-stranded DNA gaps and altered replication forks (*Abe et al., 2018*; *Fumasoni et al., 2015*; *Kouprina et al., 1992*). Ctf4 is essential for viability in vertebrates (*Abe et al., 2018*; *Yoshizawa-Sugata and Masai, 2009*), insects (*Gosnell and Christensen, 2011*), and some fungi (*Harris and Hamer, 1995*; *Williams and McIntosh, 2002*) but cannot be detected in prokaryotes, where there is a direct physical linkage between the primase (DnaG) and the helicase (DnaB) (*Lu et al., 1996*).

We generated *ctf4Δ* and wild type (WT) ancestor strains by sporulating a heterozygous *CTF4/ ctf4Δ* diploid. As previously reported (*Kouprina et al., 1992*; *Miles and Formosa, 1992*), *ctf4Δ* cells display severe growth defects, which we quantified as a fitness decrease of approximately 25% relative to WT (*Figure 1C*). We then evolved eight parallel populations of each genotype for 1000 generations by serial dilutions in rich media, freezing population samples every 50 generations (*Figure 1B*). Under this regime, spontaneous mutations that increase cellular fitness and survive genetic drift will be selected and spread asexually within the populations (*Jerison and Desai, 2015*; *Levy et al., 2015*; *Venkataram et al., 2016*). At the end of the experiment, we asked whether cells had recovered from the fitness decrease induced by replication stress by measuring the fitness of the evolved *ctf4Δ* and WT populations. Expressing the results as a percentage of the fitness of the WT ancestor, the evolved WT populations increased their fitness by an average of $4.0 \pm 0.3\%$ (*Figure 1—figure supplement 1*), a level similar to previous experiments (*Buskirk et al., 2017*; *Lang et al., 2013*). In contrast, we found that the fitness of the evolved *ctf4Δ* populations rose by $17 \pm 0.2\%$ (*Figure 1—figure supplement 1*). Clones isolated from these populations showed similar fitness increases (*Figure 1C*).

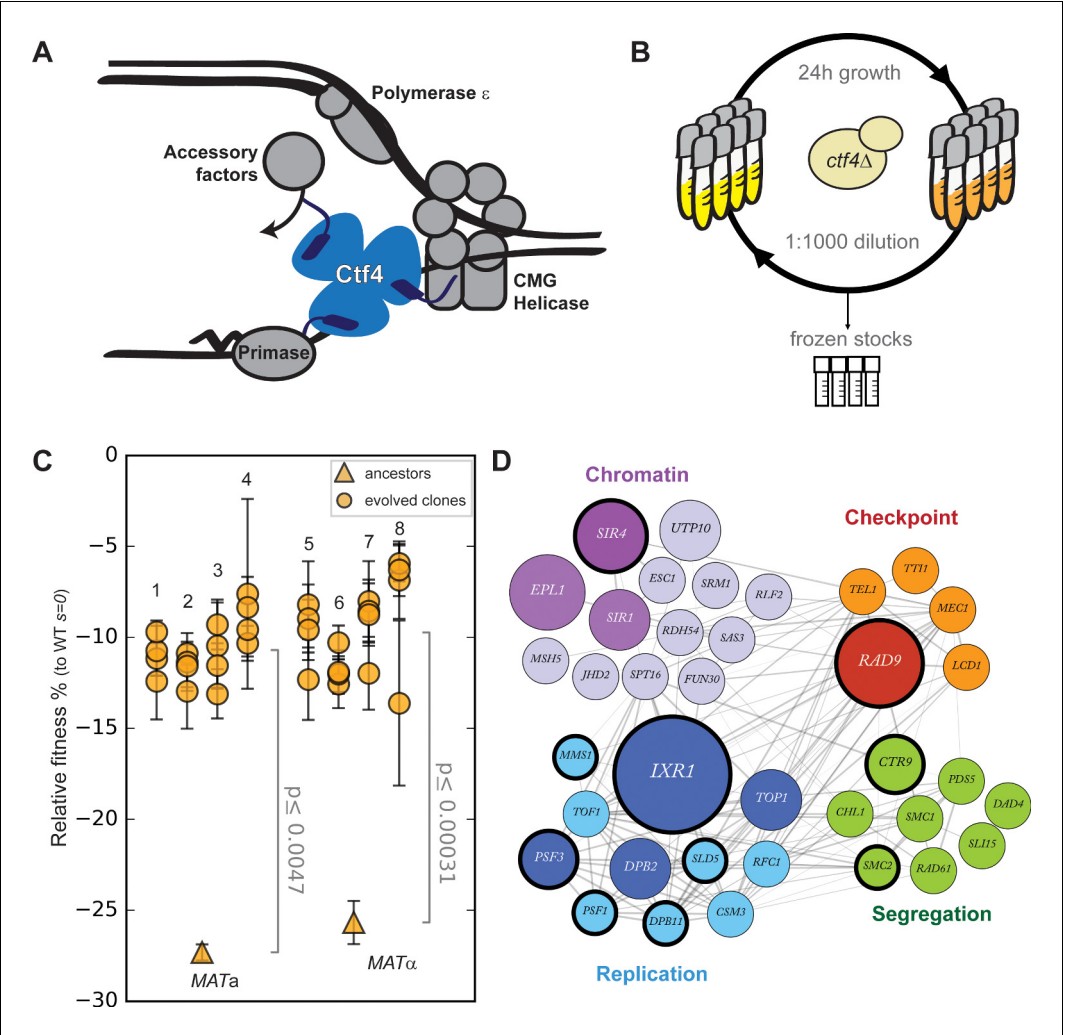

**Figure 1.** Fast evolutionary adaptation to DNA replication stress. (**A**) Schematic representation of the replisome focused on the role of Ctf4 in coordinating the replicative helicase, primase, and other factors. (**B**) The experimental evolution scheme: independent colonies of *ctf4Δ S. cerevisiae* were inoculated in rich media, grown to saturation, and diluted 1:1000 in fresh media for a total of 100 cycles (1000 generations). Populations samples were saved every 50 generations for future analysis. (**C**) Fitness of the *ctf4Δ* ancestor strains and of 32 evolved clones isolated from the 8 (labeled 1 through 8) populations derived from them, relative to wt cells (*s* = 0). Error bars represent standard deviations. *MATa* and *MATα* refer to the strain sex. (**D**) Simplified representation of the modules enriched in putative adaptive mutations, found in evolved clones. Gray lines represent evidence of genetic and physical interactions from the literature (https://string-db.org). Node diameter is proportional to the number of populations in which the gene was mutated. Selection on darker nodes was statistically significant. Nodes surrounded with a bold circle are genes in which mutations were found to strongly correlate with the evolved phenotype by bulk segregant analysis.

The online version of this article includes the following source data and figure supplement(s) for figure 1:

**Source data 1.** Numerical values corresponding to the graph in panel C.
**Figure supplement 1.** Fitness of the evolved populations.
**Figure supplement 1—source data 1.** Numerical values corresponding to the graph.
**Figure supplement 2.** Bulk segregant analysis of evolved clones.
**Figure supplement 3.** Chromosome metabolism-associated GO terms.

To understand this evolutionarily rapid adaptation to constitutive replication stress, we whole-genome sequenced all the final evolved populations as well as 32 individual clones (four from each of the evolved populations) isolated from the *ctf4Δ* lineages. During experimental evolution, asexual

populations accumulate two types of mutations: adaptive mutations that increase their fitness and neutral or possibly mildly deleterious mutations that hitchhike with the adaptive mutations (*Supplementary file 1*). To distinguish between these mutations, we used a combination of statistical and experimental approaches. First, we inferred that mutations in a gene were adaptive if the gene was mutated more frequently than expected by chance across our parallel and independent populations (*Supplementary file 2*). Second, we performed bulk segregant analysis on selected evolved clones. This technique takes advantage of sexual reproduction, followed by selection, to separate causal and hitchhiking mutations. In this case, mutations that segregate strongly with the evolved phenotype are assumed to be adaptive (*Figure 1—figure supplement 2*). We combined these two lists of mutated genes and looked for enriched gene ontology (GO) terms. This analysis revealed an enrichment of genes implicated in several aspect of chromosome metabolism (*Supplementary file 3*). Among the genes associated with these terms, many are involved in four functional modules: DNA replication, chromosome segregation (including genes involved in sister chromatid linkage and spindle function), cell cycle checkpoint and chromatin remodeling (*Figure 1— figure supplement 3*). The genes in these modules that were mutated in the evolved clones are shown, grouped by function, in *Figure 1D*.

## DNA replication stress selects for inactivation of the DNA damage checkpoint

We found several mutations affecting genes involved in cell-cycle checkpoints (*Figure 2B*). Checkpoints are feedback control mechanisms that induce cell-cycle delays in response to defects that reflect the failure to complete important process and thus guarantee the proper sequence of events required for cell division (*Elledge, 1996*; *Murray, 1992*). Three delays, caused by DNA damage or defects in DNA replication, have been characterized. The first prevent cells from entering S-phase in response to DNA damage occurring in G1. A second slows progress through S-phase in response to problems encountered during DNA synthesis. The third delays sister chromatid separation (anaphase) and the exit from mitosis in response to DNA damage incurred after cells enter S-phase (*Figure 2A*; *Murray, 1994*).

The genes listed in *Figure 2B* are implicated at different levels in either the replication or mitotic delays (*Figure 2—figure supplement 1B*; *Pardo et al., 2017*). The most frequently mutated gene, *RAD9*, encodes an important component of the DNA damage checkpoint, which is required to slow DNA synthesis and delay anaphase in response to DNA lesions (*Weinert and Hartwell, 1988*). Four out of the five mutations in *RAD9* produced early stop codons, or radical amino acid substitutions in the BRCT domain, which is essential for Rad9's function (*Figure 2C*, *Figure 2—figure supplement 1A*; *Soulier and Lowndes, 1999*), arguing that inactivation of Rad9 was repeatedly and independently selected for during evolution. To test this hypothesis, we engineered the most frequently occurring mutation (*2628 +A*, a frameshift mutation leading to a premature stop codon K883*) into the ancestral *ctf4Δ* strain (*ctf4Δ* anc). We suspect that the high frequency of this mutation is due to the presence of a run of 11 As, a sequence that is known to be susceptible to loss or gain of a base during DNA replication. This mutation (*Figure 2C*, *Figure 2—figure supplement 1A*) produced a fitness increase very similar to the one caused by deleting the entire gene (*Figure 2D*). We conclude that inactivation of Rad9 is adaptive in the absence of Ctf4.

We asked if the removal of Rad9 eliminated a cell cycle delay caused by the absence of Ctf4. In the *ctf4Δ* ancestor, *rad9Δ* does, indeed, decrease the fraction of cells with a 2C DNA content (the DNA content in G2 and mitosis) observed in asynchronously growing *ctf4Δ* cells (*Tanaka et al., 2009*). This observation suggests that the interval between the end of DNA replication and cell division decreases in *ctf4Δ rad9Δ* cells. The spindle checkpoint, which blocks anaphase in response to defects in mitotic spindle assembly, can also delay chromosome segregation in cells (*Li and Murray, 1991*). But although deleting *MAD2*, a key spindle checkpoint component, also decreases the interval between replication and division in *ctf4Δ* cells (*Hanna et al., 2001*), it reduces rather than increases the fitness of the *ctf4Δ* ancestor (*Figure 2D*). These results suggest that ignoring some defects in *ctf4Δ* cells, such as those that activate the DNA damage checkpoint, improves fitness, whereas ignoring others, such as defects in chromosome alignment on the spindle, reduces fitness.

Problems encountered during DNA synthesis also activate the replication checkpoint, which inhibits DNA replication to prevent further lesions (*Zegerman and Diffley, 2009*; *Zegerman and Diffley, 2010*). As many proteins involved in the DNA damage checkpoint are shared with the replication

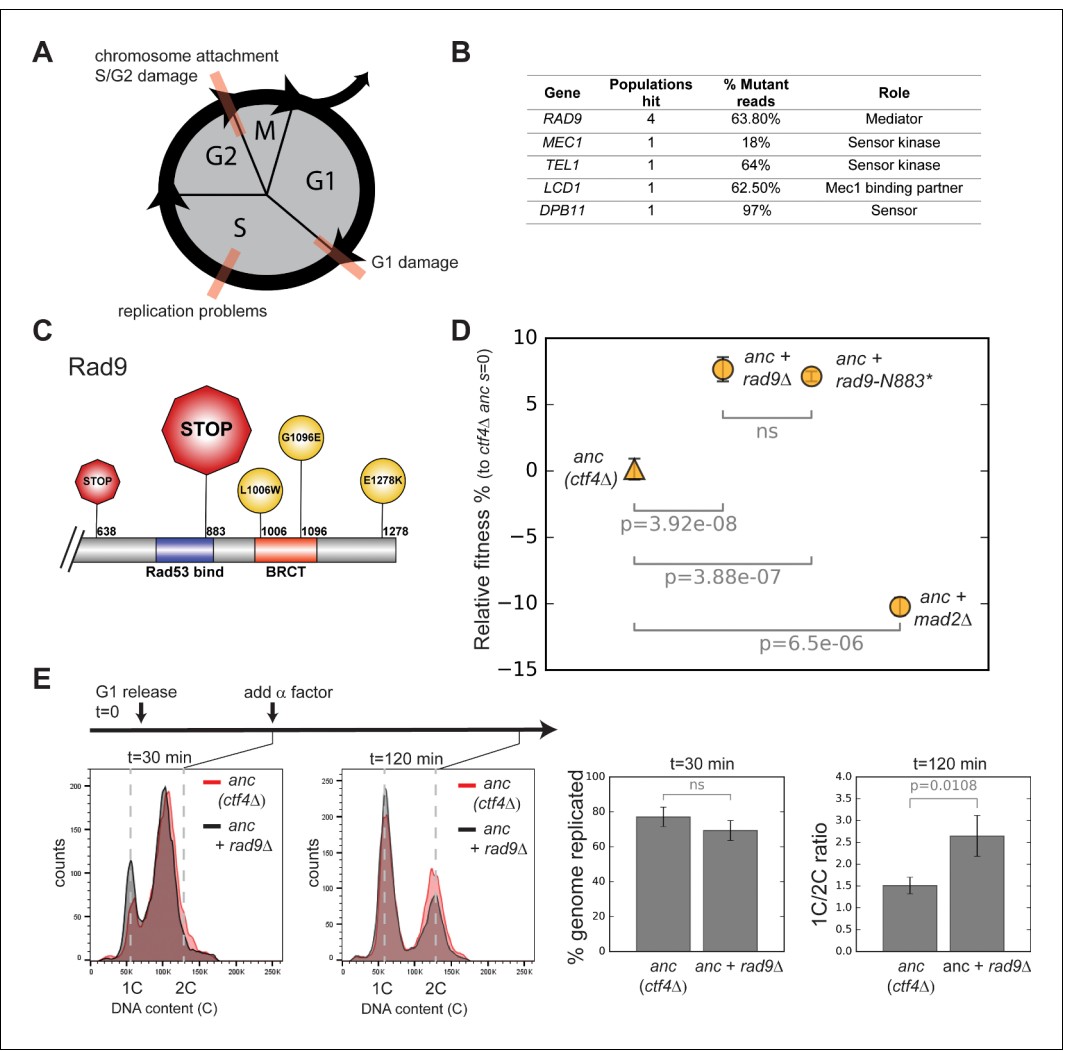

**Figure 2.** Checkpoint mutations cause a faster G2/M transition in evolved cells. (**A**) Schematic representation of cell cycle progression. The transitions delayed by various checkpoints are highlighted in red. (**B**) List of checkpoint genes mutated in evolved clones and their role in the signaling cascade. 'Populations hit' refers to the number of populations where the gene was mutated. '% Mutant reads' was calculated as the average of the mutant read frequencies in the different populations where the mutation was detected. (**C**) Schematic of the C-terminal region of Rad9 that was affected by mutations in evolved clones. The diameter of the symbol is proportional to the number of populations where the mutation was detected. Note that both stop codons resulted from an upstream frameshift. Two populations contained more than one distinct *RAD9* mutations. (**D**) The fitness of *ctf4Δ* strains carrying two reconstructed mutations in the DNA damage checkpoint (*rad9Δ* and *rad9K883\**) and an engineered inactivation of the spindle checkpoint (*mad2Δ*) relative to the *ctf4Δ* ancestors (*ctf4Δ anc, s=0*). Error bars represent standard deviations. (**E**) Cell cycle profiles of *ctf4Δ* ancestor and *ctf4Δ rad9Δ* cells at two time points during a synchronous cell cycle. Cells were arrested in G1 and subsequently released synchronously into S-phase. Time points taken at 30 min and 120 min after the release are shown. 1C is the DNA content of a cell in G1. α-factor was added 30 min after release to prevent cells entering a second cell cycle and thus ensure that 2C cells at the 120 min measurement resulted from a G2 delay rather than progress through a second cell cycle. The percentage of genome replicated at 30 min was calculated based on the cell cycle profile. 1C/2C ratios were calculated based on the height of the respective 1C and 2C peaks at 120 min.

The online version of this article includes the following source data and figure supplement(s) for figure 2:

**Source data 1.** Numerical values corresponding to the graph in panel D.
**Figure supplement 1.** Mutations in checkpoint genes.

checkpoint (*Figure 2—figure supplement 1B*; *Pardo et al., 2017*), we followed a single synchronous cell-cycle to ask whether the fitness benefits conferred by *RAD9* deletion were due to a faster progression through S-phase or faster progress through mitosis. Loss of Rad9 in *ctf4Δ* cells did not accelerate S-phase, but it did lead to faster passage through mitosis as revealed by a reduced fraction of 2 C cells (*Figure 2E*).

To separate the role of the replication and DNA damage checkpoints, we genetically manipulated targets of the checkpoints whose phosphorylation delays either anaphase (Pds1, *Wang et al., 2001*) or the completion of replication (Sld3 and Dbf4, *Zegerman and Diffley, 2010*, *Figure 2—figure supplement 1B*). Fitness measurement in these mutants (*pds1-m9* or the double mutant *sld3-A/ dbf4-4A*) showed that while decreasing the mitotic delay in ancestral *ctf4Δ* cells was beneficial, a faster S-phase was highly detrimental (*Figure 2—figure supplement 1C*). Collectively, these results show that the specific absence of a DNA damage-induced delay of anaphase, rather than generic cell-cycle acceleration, is adaptive in *ctf4Δ* cells experiencing replication stress.

## Amplification of cohesin loader genes improves sister chromatid cohesion

We examined the evolved clones for changes in the copy number across the genome (DNA copy number variations, CNVs). Several clones showed segmental amplifications, defined as an increase in the copy number of a defined chromosomal segment (*Figure 3—figure supplement 1*). The most common CNV in evolved *ctf4Δ* cells (17 out of 32 sequenced clones) was the amplification of a 50–100 kb region of chromosome IV (chrIV). In addition to this segmental amplification, evolved clone EVO2-10 also carried an extra copy of a portion of chromosome V (chrV, *Figure 3A*). The eight evolved wild type populations had no segmental amplifications, suggesting that changes in copy number were a specific adaptation to constitutive DNA replication stress.

Amongst the genes affected by these two CNVs are *SCC2* and *SCC4*, on the amplified portions of chromosomes IV and V respectively. These two genes encode the two subunits of the cohesin loader complex, which loads cohesin rings on chromosomes to ensure sister chromatid cohesion until anaphase (*Figure 3—figure supplement 2B*; *Ciosk et al., 2000*; *Michaelis et al., 1997*). The amplification of *SCC2* and *SCC4*, together with the other genes altered by point mutations in our evolved clones (*Figure 3B*, *Figure 3—figure supplement 2A*), strongly suggest that the absence of Ctf4 selects for mutations that affect the linkage between sister chromatids.

*CTF4* was originally identified because mutants in this gene reduced the fidelity of chromosome transmission (CTF = chromosome transmission fidelity, *Spencer et al., 1990*); later studies showed that this defect was due to premature sister chromatid separation, which resulted in increased chromosome loss at cell division (*Hanna et al., 2001*). We hypothesized that the segmental amplifications of chrIV and chrV were selected to increase the amount of the cohesin loading complex. To test this idea, we reintroduced a second copy of these genes in a *ctf4Δ* ancestor. As predicted by the more frequent amplification of *SCC2*, we found that while an extra copy of *SCC4* alone did not significantly affect fitness, an extra copy of *SCC2*, or an extra copy of both *SCC2* and *SCC4* increased fitness by 4–5% (*Figure 3C*). Consistent with a role of *SCC4* amplification only in combination of *SCC2* amplification, we found that the segmental amplification of chrV followed that of chrIV in the EVO2 population (*Figure 3—figure supplement 3A–B*). We examined cells arrested in mitosis to measure the extent of premature sister chromatid separation in the same strains. Adding extra copies of the cohesin loader subunits improved sister chromatid cohesion (*Figure 3D*) and the amplitude of the improvement in sister cohesion for different strains had the same rank order as their increase in fitness (*Figure 3C*). We conclude that the increased copy number of the cohesin loader subunits is adaptive and alleviates the cohesion defects induced by the lack of Ctf4.

## Altered replication dynamics promote DNA synthesis in late replication zones

We found mutations in several genes involved in DNA replication (*Figure 4A*, *Figure 4—figure supplement 1A*). Among these, we found four independent mutations (*Figure 4B*) that altered three different subunits of the replicative CMG (Cdc45, MCM, GINS) helicase (*Labib and Gambus, 2007*; *Moyer et al., 2006*). The CMG helicase is bound in vivo by Ctf4 through the GINS subunit Sld5 (*Simon et al., 2014*). This binding allows Ctf4 to coordinate the helicase's progression with primase,

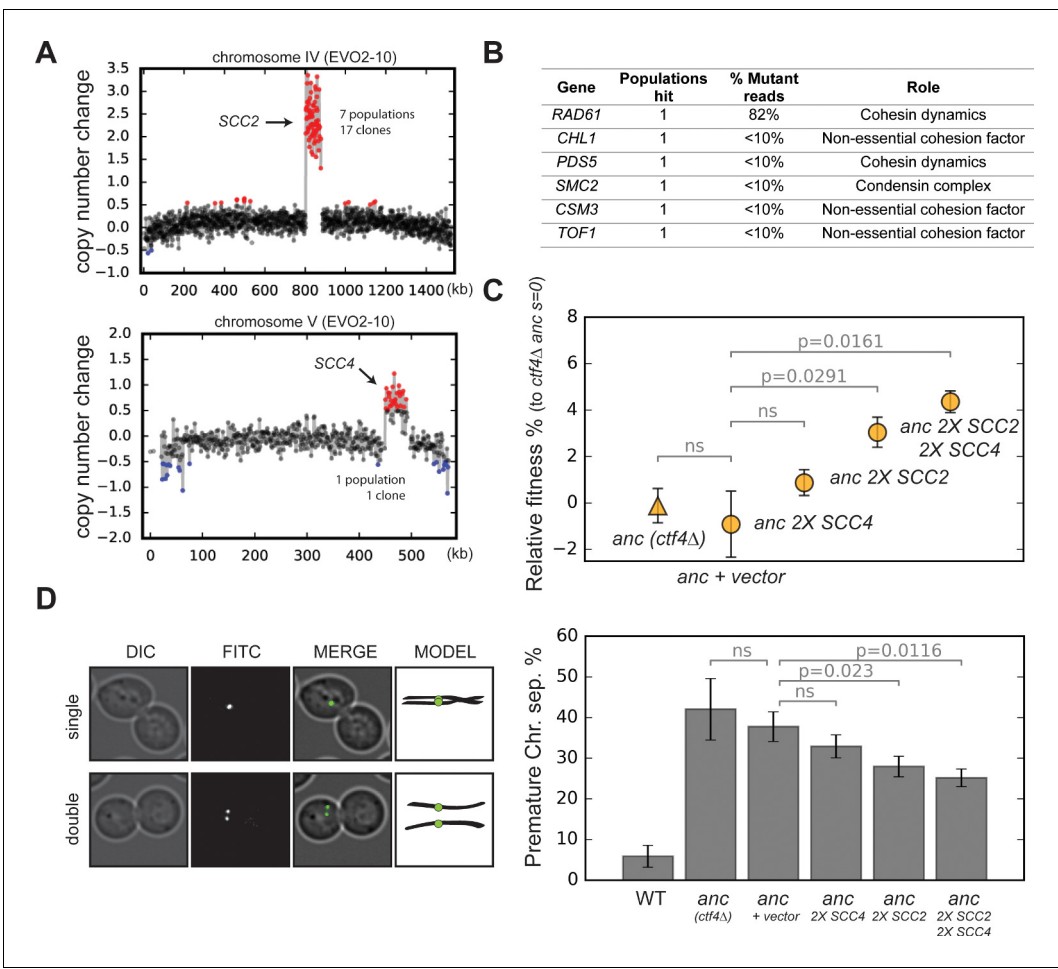

**Figure 3.** Amplification of cohesin loader genes. (**A**) Copy number variations (CNVs) affecting chromosome IV and chromosome V in clone EVO2-10. Copy number change refers to the fragment's gain or loss during the evolution experiment (i.e. +1 means that one copy was gained). Red highlights gains, blue highlights losses. (**B**) List of genes involved in chromosome segregation that were mutated in evolved clones, and their respective role in the process. 'populations hit' is the number of populations where the gene was found mutated. '% Mutant reads' was calculated as the average of the mutant read frequencies in the different populations where the mutation was detected. (**C**) Fitness of ancestral, *ctf4Δ* strains that carry chromosomally integrated extra copies of cohesin loader genes, relative to the *ctf4Δ* ancestor (*s* = 0). Error bars represent standard deviations. (**D**) Premature chromatid separation assay: Cells which contained a chromosome marked by a GFP dot (Lac repressor-GFP binding to an array of LacO sites) were arrested in metaphase and visualized under the microscope. The number of dots reports on premature sister chromatid separation. Two sister chromatids that are still linked to each other produce a single fluorescent dot (single, left panel), while cells whose sister chromatids have separated contain two distinguishable dots (double, left panel). Quantitation of premature sister chromatid separation in cells carrying extra copies of cohesin loader genes (right panel).

The online version of this article includes the following source data and figure supplement(s) for figure 3:

**Source data 1.** Numerical values corresponding to the graph in panel C.
**Source data 2.** Numerical values corresponding to the graph in panel D.
**Figure supplement 1.** CNVs affecting the genome of the 32 isolated evolved clones.
**Figure supplement 2.** Mutations affecting chromosome segregation.
**Figure supplement 3.** *SCC2/4* amplification in EVO2.

which synthesizes the primers for lagging strand DNA synthesis, and other factors recruited behind the replication fork (*Figure 1A*; *Samora et al., 2016*; *Villa et al., 2016*). A CMG helicase mutation found in one of the evolved clones, *sld5-E130K*, increased the fitness of the ancestral *ctf4Δ* strain (*Figure 4C*).

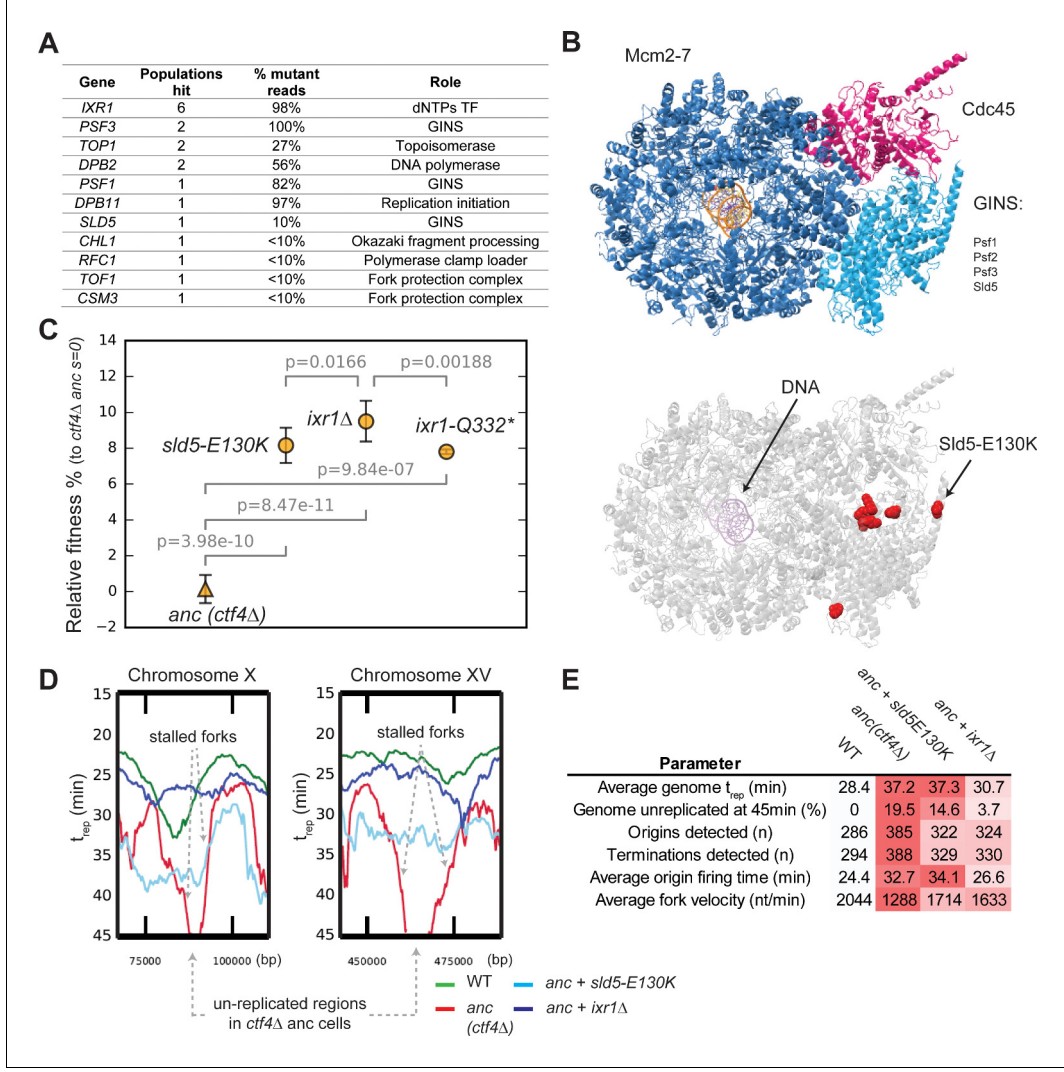

**Figure 4.** Adaptive mutations change DNA replication dynamics. (**A**) Genes involved in DNA replication that were mutated in evolved clones, and their role in replication. 'populations hit' is the number of populations where the gene was found mutated. '% Mutant reads' was calculated as the average of the mutant read frequencies in the different populations where the mutation was detected. (**B**) Structure of the CMG helicase (PDB:5u8s, upper panel) highlighting the catalytic subunits (Mcm2-7) and the regulatory subunits (Cdc45 and GINS). Red spheres represent the residues affected by mutations found in evolved clones (lower panel). (**C**) The fitness of $ctf4\Delta$ strains carrying reconstructed mutations in the replicative helicase ($sld5$-$E129K$) and in $IXR1$ ($ixr1\Delta$ and $ixr1$-$Q332*$) relative to the $ctf4\Delta$ ancestor ($s = 0$). Error bars represent standard deviations. (**D**) DNA replication profiles: cells were arrested in G1 and released into a synchronous S-phase, taking samples every 15 min for whole genome sequencing analysis. Change in DNA copy number over time were analyzed and used to calculate $t_{rep}$ (time at which 50% of the cells in the population have replicated a given region (***Figure 4—figure supplement 3***, see material and methods for details). Snapshots of regions from chromosome X and XV are shown as examples, highlighting the presence of stalled forks and unreplicated regions in $ctf4\Delta$ cells (which are absent in strains that also carry $sld5$-$E130K$ or $ixr1\Delta$ mutations). (**E**) Quantitative analysis of DNA replication. Each parameter was derived from the genome-wide DNA replication profile of each sample (***Figure 4—figure supplement 3***, see material and methods for details). Heatmaps refer to the severity of the defect (white = wt, red = ctf4Δ ancestor).

The online version of this article includes the following source data and figure supplement(s) for figure 4:

**Source data 1.** Numerical values corresponding to the graph in panel C.
**Figure supplement 1.** Mutations affecting DNA replication.
**Figure supplement 2.** Forks progression in DNA replication profiles.
**Figure supplement 3.** DNA replication profiles.
**Figure supplement 4.** Effect of helicase mutations and dNTPs levels on fitness.
*Figure 4 continued on next page*

*Figure 4 continued*

**Figure supplement 4—source data 1.** Numerical values corresponding to the graph in panel A.
**Figure supplement 4—source data 2.** Numerical values corresponding to the graph in panel B.

*IXR1*, a gene indirectly linked to DNA replication, was mutated in several populations (*Figure 4A*). *IXR1* encodes for a transcription factor that indirectly and positively regulates the concentration of deoxyribonucleotide triphosphates (dNTPs, *Tsaponina et al., 2011*), the precursors for DNA synthesis. The occurrence of multiple nonsense mutations in this gene strongly suggested selection to inactivate Ixr1 (*Figure 4—figure supplement 1B*). Consistent with this prediction, we found that engineering either a nonsense mutation (*ixr1-Q332\**) or a gene deletion conferred a selective advantage to *ctf4Δ* ancestor cells (*Figure 4C*).

We asked how mutations in the replicative helicase or inactivation of *IXR1* increased the fitness of *ctf4Δ* cells. One hypothesis is that the absence of Ctf4 reduces the coordination of activities required to replicate DNA and leads to the appearance of large regions of single stranded DNA, which in turn exposes the forks to the risk of nuclease cleavage or collapse. If this were true, slowing the replicative helicase or the synthesis of the leading strand would reduce the amount of single stranded DNA near the replication fork and improve the ability to complete DNA replication before cell division. To test this idea, we used whole genome sequencing at different points during a synchronous cell cycle to compare the dynamics of DNA replication (*Figure 4—figure supplement 2*) in four strains: WT, the *ctf4Δ* ancestor, and *ctf4Δ* strains containing either the *sld5-E130K* or *ixr1Δ* mutations.

We found that cells lacking Ctf4 experience several defects compared to WT: on average, origins of replication fire later and DNA replication forks proceed more slowly across replicons, often showing fork stalling (*Figure 4D–E*, *Figure 4—figure supplement 3*). As a consequence of these two defects, cells still contain significant regions of unreplicated DNA late in S-phase (45 min, *Figure 4D–E*, *Figure 4—figure supplement 3*). Both *sld5-E130K* or *ixr1Δ* mutations significantly increase the average replication fork velocity primarily by avoiding stalls in DNA replication and thus leading to earlier replication of the regions that replicate late in the ancestral *ctf4Δ* cells (*Figure 4D–E*, *Figure 4—figure supplement 3*). Altogether, these results show that cells evolved modified DNA replication dynamics to compensate for defects induced by DNA replication stress.

## Epistatic interactions among adaptive mutations dictate evolutionary trajectories

Can we explain how the ancestral *ctf4Δ* strains recovered to within 10% of WT fitness in only 1000 generations? Although all the mutations that we engineered into *ctf4Δ* ancestor cells reduce the cost of DNA replication stress, none of them, individually, account for more than a third of the fitness increase observed over the course of the entire evolution experiment (*Figure 1C*). Sequencing individual evolved clones revealed the presence of mutations in at least two of the three modules whose effects we analyzed in isolation (*Figure 5—figure supplement 1*, *Supplementary file 1*). We therefore asked if we could recapitulate the fitness of the evolved clones by adding adaptive mutations from multiple different modules to the *ctf4Δ* ancestor. We obtained all possible combinations of two, three, and four adaptive mutations, in the *ctf4Δ* ancestor, by sporulating a diploid strain that was heterozygous for all four classes of adaptive mutations: inactivation of the DNA damage checkpoint (*rad9Δ*), amplification of the cohesin loader (an extra copy of *SCC2*), alteration of the replicative helicase (*sld5-E130K*), and altered regulation of dNTP pools (*ixr1Δ*).

We found that the two mutations that affected DNA replication were negatively epistatic (*Figure 5A*): in the presence of *ctf4Δ*, strains that contained both *sld5-E130K* and *ixr1Δ* were not significantly more fit than strains that contained only *ixr1Δ* and the quadruple mutant (*2X-SCC*, *rad9Δ*, *sld5-E130K*, *ixr1Δ*) was much less fit than the two triple mutants that contained only one of the two mutations that affected DNA replication (*2X-SCC*, *rad9Δ*, *sld5-E130K* and *2X-SCC*, *rad9Δ*, *ixr1Δ*). As a result, the two fittest strains carry only three mutations: in both cases, they affected the three modules we previously characterized: sister chromatid linkage and chromosome segregation (*2X-SCC2*), the DNA damage checkpoint (*rad9Δ*) and DNA replication (*sld5-E130K* or *ixr1Δ*). These two strains

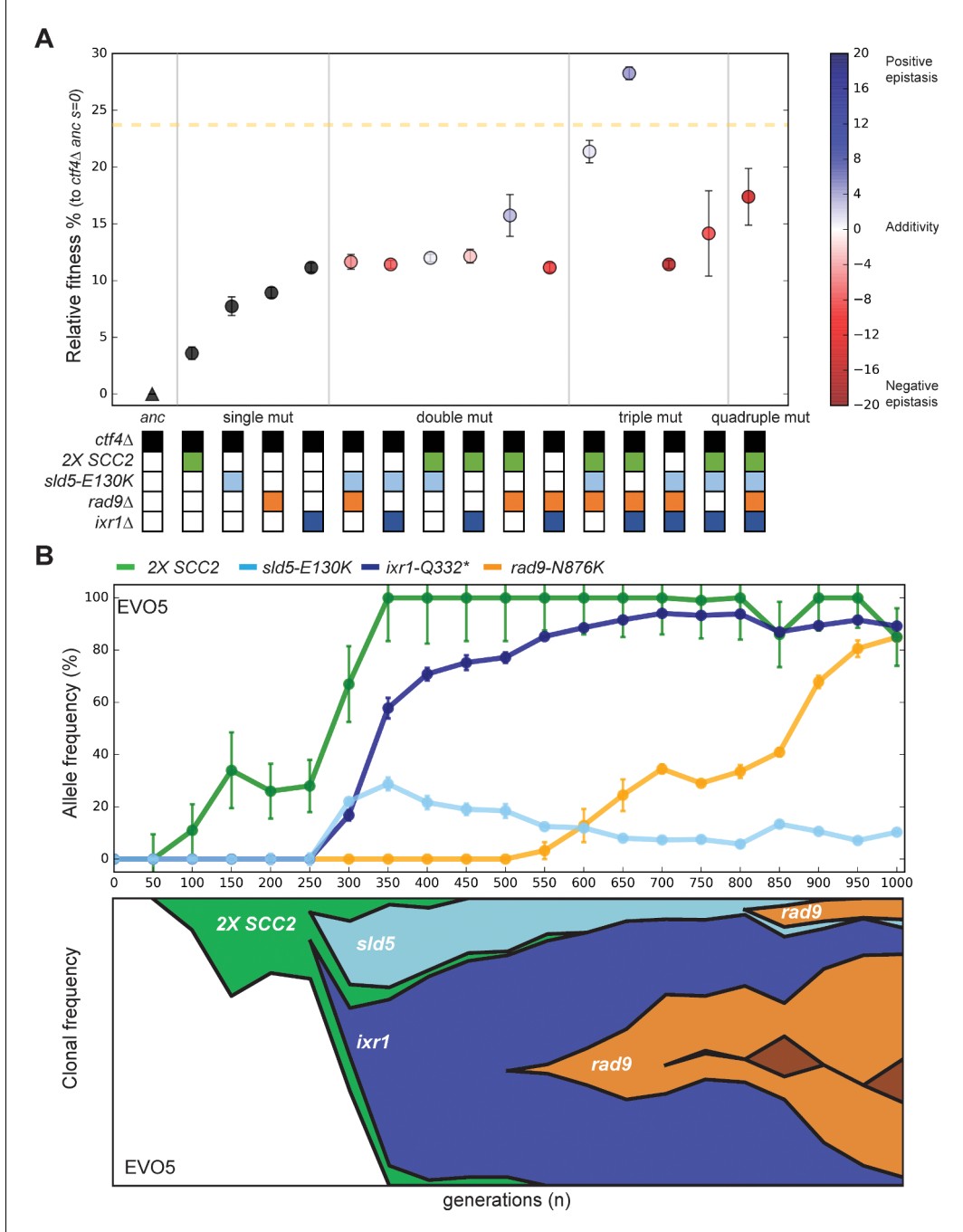

**Figure 5.** Epistatic interaction and evolutionary dynamics. (A) Fitness of all possible combinations of four adaptive mutations in the *ctf4Δ* ancestral background. The fitness measurements are relative to *ctf4Δ* ancestors (*s* = 0). The dashed yellow line represents the average fitness of clones isolated from EVO5. Note that, unlike *Figure 1C*, fitness values are calculated relative to the ancestral *ctf4Δ* strain, and not to WT (hence the differences in absolute values, see material and methods). Error bars represent standard deviations. The fitnesses of individual strains are colored using the heatmap to the right of the figure, which represents epistasis: white = perfect additivity, red = negative epistasis (antagonism), blue = positive epistasis (synergy). Colors in the heatmap represent the deviation in percentage between the observed fitness and the one calculated by adding the fitness effects of the individual mutations. (B) The temporal spread of mutant alleles during the experimental evolution of population EVO5 (upper panel). Error bars represent standard deviations. Genomic DNA was extracted from population samples, mutated loci were PCR amplified and Sanger sequencing was used to measure allele ratios (upper panel). A Muller diagram representing the lineages evolving in population EVO5 (lower panel). Data was obtained

*Figure 5 continued on next page*

*Figure 5 continued*

by combining alleles frequencies with their linkage as revealed by whole genome sequencing of clones isolated from EVO5 (*Figure 5—figure supplement 2* and *Supplementary file 1*).

The online version of this article includes the following source data and figure supplement(s) for figure 5:

**Source data 1.** Numerical values corresponding to the graph in panel A.
**Source data 2.** Numerical values corresponding to the graph in panel B (upper panel).
**Source data 3.** Numerical values corresponding to the graph in panel B (lower panel).
**Figure supplement 1.** Reproducibility in evolutionary trajectories.
**Figure supplement 1—source data 1.** Numerical values corresponding to the graphs.
**Figure supplement 2.** Evolutionary history of EVO5.

displayed a fitness comparable to the average of the evolved populations (*Figure 1C*), suggesting that we had recapitulated the major adaptive events in our engineered strains.

We asked if the antagonistic interaction between *sld5-E130K* and *ixr1Δ* seen in our reconstructed strains had also occurred in our evolution experiment. We focused on an evolved population (EVO5) that carried all the mutations described above and analyzed the allele frequency in the intermediate samples collected across the evolution experiment. By following the frequency of alleles within the population and sequencing individual clones, we found that the mutations in the three modules happened in three consecutive selective waves: first, cells acquired an extra copy of the cohesin loader-encoding gene *SCC2*, second, *ixr1-Q332\** and *sld5-E130K* appeared, simultaneously, in two different lineages, and finally *rad9-N876K* appeared independently in the two lineages containing either *ixr1-Q332\** or *sld5-E130K* (*Figure 5B*, *Figure 5—figure supplement 2*). After their initial appearance, the two lineages containing *ixr1-Q332\** or *sld5-E130K* competed with each other for the remainder of the experiment. In this population, both final lineages accumulated mutations whose interaction was nearly additive or positively epistatic and avoided combinations that show strong negative epistasis (*Figure 5A*). Thus, although negative epistasis exists, selection finds trajectories that avoid it, as previously observed in a similar experiment perturbing cell polarity (*Laan et al., 2015*).

## Discussion

Many fundamental processes in cell biology have a conserved underlying structure despite substantial variation in their detailed mechanisms, leading to curiosity about how these changes can occurred without destroying the overall process. One approach to this question is to compare processes in related organisms and use classical and molecular genetics to find the genetic variants responsible for inter-species differences. This strategy has the disadvantage that many of the mechanistic changes happened so long ago that it is difficult to exchange components between their current-day descendants. We therefore used the alternative approach of applying a physiological stress that reduced the fitness of an organism and using experimental evolution to accumulate, identify, and study the mutations that increase fitness and adapt the organism to the stress. Using this approach allowed us to ask several questions about the evolution of conserved processes: i) how rapidly and how completely does fitness increase after a severe perturbation, ii) how reproducible are the evolutionary trajectories of replicate populations, iii) what genes are mutated, and which function modules do they affect, iv) what are the molecular mechanisms of adaptation, v) how do distinct mechanisms interact with each other, and vi) what do the mechanistic changes reveal about the evolutionary plasticity of the perturbed process.

To investigate the evolution of conserved cellular processes, we studied the evolutionary adaptation of cells experiencing constitutive DNA replication stress induced by the lack of a protein, Ctf4, that plays an important role in DNA replication. We tested whether significant changes in DNA replication could be acquired as a consequence of constitutive DNA replication stress. We found that over 1000 generations, populations increased from 75% to 90% of the fitness of their wild-type ancestors by sequentially accumulating mutations that individually affect three different functions that contribute to chromosome metabolism: DNA replication, chromosome segregation and the DNA damage checkpoint.

Our experiment reveals the short-term evolutionary plasticity of chromosome metabolism. A single genetic perturbation and a thousand generations are enough to select for significant changes in

three functional modules affecting chromosome metabolism. By the end of the experiment, evolved lineages had sequentially modified chromosome cohesion, changed the dynamics of DNA replication, and lost an important cell-cycle response to DNA damage. These changes combine to produce the evolved phenotype and allow cells to approach wild type fitness despite the presence of continued DNA replication stress. This result suggests that despite their conservation, these modules and the connections between them are evolutionarily plastic and can accommodate short-term responses to strong perturbations, helping to explain differences that have accumulated over hundreds to billions of years of evolution.

Previous studies have argued that perturbations in DNA replication are less likely to be repaired by single compensatory mutations than other processes, such as intracellular trafficking (*Liu et al., 2015*; *van Leeuwen et al., 2016*). We believe that the explanation for the difference between these studies and our own lies in the different nature of the mutations that are selected. Suppressor screens rely on single mutations that can either rescue lethality or whose fitness effect is greater than the noise in systematic analyses of genetic interactions. In contrast, experimental evolution following non-lethal perturbations allows for the sequential acquisitions of small-effect mutations that collectively rescue the perturbed process. Our experiment suggests that although the single mutations that can fully repair genetic damage to DNA replication are rare, the existence of combination of small-effect mutations that can repair perturbations make it evolutionarily plastic.

We argue that coupling a genetic dissection of the mutations that increase fitness to a cell biological understanding of their mechanism is essential to reaching a comprehensive understanding of evolutionary change. We discuss the molecular mechanisms of adaptation, then consider how they interact to produce the final evolved phenotype, and close by commenting on the implications of our results for natural populations and cancer.

## Molecular insights into evolutionary adaptation

Cells lacking Ctf4 show an increased frequency of chromosome mis-segregation due to premature sister chromatid separation, but the mechanism underlying this defect is still unclear. Seven of our eight populations amplified *SCC2*, which encodes for one of the subunits of the cohesin loader complex. The simplest explanation for this result is that, the absence of Ctf4 restricts the productive loading of cohesin molecules that establish the linkage between sister chromatids. We propose that amplifying the genes for the cohesin loader would increase its expression, increase the productive cohesin loading and improve the linkage between sister chromatids. Improving sister chromatid cohesion allows the evolved cells to segregate their chromosomes more accurately at mitosis, avoiding mitotic delays due to the spindle checkpoint, decreasing cell death and increasing fitness (*Figure 6A*).

Persistent, cohesin-independent linkages between sister chromatids are an alternative source of segregation errors. These links include unreplicated regions of DNA or un-resolved recombination structures (*Ait Saada et al., 2017*; *Chan et al., 2007*). If they persist after the removal of cohesin, they become lingering physical links (anaphase bridges) between sister chromatids that can lead to chromosome breakage or mis-segregation during anaphase (*Chan et al., 2009*; *Gisselsson et al., 2000*). Avoiding these problems requires that replication origins fire efficiently and replication forks move continuously. Our analysis of the dynamics of DNA replication argues that a combination of frequent fork stalling and slower origin firing causes under-replication of certain chromosomal regions in the ancestral *ctf4Δ* cells. We found that severe fork stalling in *ctf4Δ* cells frequently occurs near tRNA genes, Long Terminal Repeats (LTRs) and transposable elements (Ty) (*Supplementary file 4*). These chromosomal features were previously found to be associated with replication pausing sites (*Deshpande and Newlon, 1996*; *Gadaleta and Noguchi, 2017*; *Zaratiegui et al., 2011*), suggesting that the absence of Ctf4 may exacerbate the problems in replicating these regions. We propose that these defects selected for mutations that stabilize the replication forks, allowing the timely completion of genome replication. We speculate that these mutations have the apparently paradoxical effect of accelerating DNA replication by slowing down the replication forks: mutations like *sld5-E130K* and *ixr1Δ* may slow helicase progression, stabilizing the forks by preventing frequent fork stalling or collapse and producing a higher overall fork velocity (*Figure 6B*). This hypothesis is consistent with two observations: first, although the *sld5* mutation is beneficial in ancestor cells, it decreases the fitness of WT cells (*Figure 4—figure supplement 4A*), a result we would expect from a slower replicative helicase. Second, reduced dNTPs concentrations

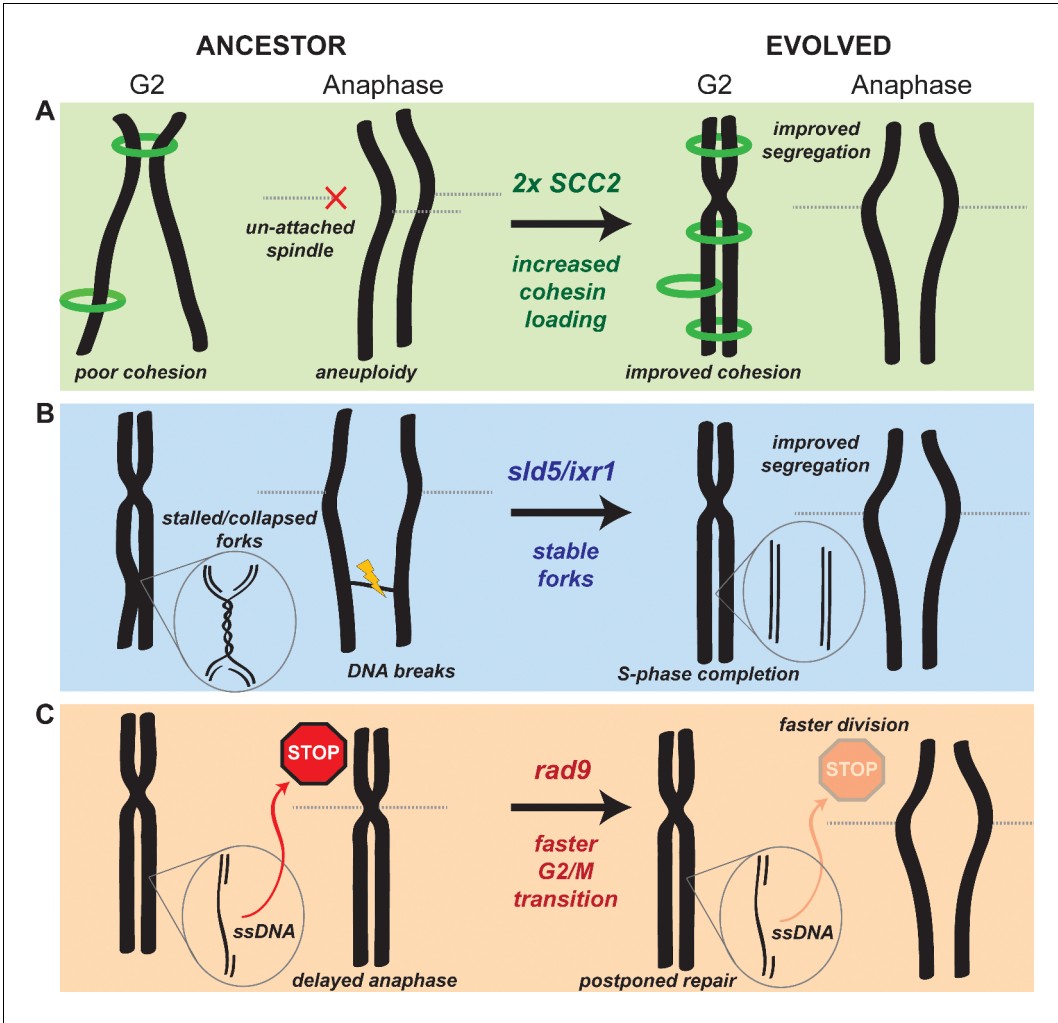

**Figure 6.** Mechanistic models of adaptation. (**A**) Amplification of the cohesin loader subunit *SCC2* increases cohesin loading and sister chromatid cohesion leading to accurate chromosome segregation (**B**) Mutations of the replicative helicase (*sld5*) or in *ixr1* stabilize replication forks and ensure the completion of chromosome replication before anaphase. (**C**) Mutations in *rad9* abolish the DNA damage checkpoint response triggered by stretches of single strand DNA (ssDNA) and allow faster cell division.

reduce fork speed by slowing polymerase incorporation rates (*Koren et al., 2010*; *Pai and Kearsey, 2017*; *Poli et al., 2012*) and inactivating Ixr1 reduces dNTP concentrations (*Tsaponina et al., 2011*). We tested this prediction by using an experimental system to manipulate dNTP concentrations: decreasing dNTP concentrations increased the fitness of *ctf4Δ* cells, while inducing higher dNTP production reduced fitness (*Figure 4—figure supplement 4B*).

Our evolved populations also accumulated mutations that inactivated the DNA damage checkpoint (*Figure 2B–D*). The benefit of these mutations arises from the loss of the DNA damage checkpoint's ability to delay the start of anaphase (*Figure 2E*, *Figure 2—figure supplement 1B*). The absence of Ctf4 induces aberrant DNA structures and ssDNA that induce moderate activation of the checkpoint (*Poli et al., 2012*), which delays the start of anaphase, increasing doubling time and thus decreasing fitness (*Figure 2D–E*). Inactivating Rad9 eliminates the delay, shortening the time required for mitosis and increasing fitness (*Figure 2E*, *Figure 2—figure supplement 1C*, *Figure 6C*).

This solution seems counter-intuitive, as the loss of a safeguard mechanism such as the DNA damage checkpoint should cause genetic instability in cells suffering from replication stress. The resolution of this paradox may lie in the overlapping action of the replication, DNA damage, and spindle

checkpoints. We propose that the replication and the spindle checkpoints delay the cell cycle in response to defects that would kill the ancestral *ctf4Δ* cells, such as excessive replication fork collapses and pairs of sister chromatids attached to the same spindle pole, whereas the damage checkpoint responds to defects, like regions of single-stranded DNA, that can be repaired after cell division.

## Epistatic interactions between adaptive mechanisms

We asked how the mutants we identified and analyzed interacted with each other and whether they could explain the fitness of our evolved populations. Measuring allele frequencies over time and engineering all possible combinations of adaptive mutations allowed us to propose a detailed model for the evolutionary trajectories of our population 5 (EVO5). Segmental amplifications form at a higher frequency than other types of mutation (*Lynch et al., 2008*; *Sharp et al., 2018*; *Yona et al., 2015*); although most are detrimental, the amplification of specific genes can be advantageous and cause rapid adaptation (*Adamo et al., 2012*; *Gresham et al., 2008*; *Hughes et al., 2000*; *Payen et al., 2014*). Thus, the first event in EVO5 is the spread of a segmental amplification of chromosome IV containing *SCC2*, which improves fitness by reducing cohesion defects. In this lineage, mutations in the replicative helicase, *sld5-E130K*, and *ixr1-Q332\** were then detected almost simultaneously but in different clones. Above, we suggest that both mutations slow replication forks. If there is an optimal fork speed in *ctf4Δ* cells, the presence of a second mutation of this class might be ineffective or even detrimental if the forks move too slowly, explaining the negative epistasis we observed. Because the *ixr1* and *sld5* mutations improve DNA replication to a similar extent, the two lineages have comparable fitness, explaining the clonal interference that persists for the rest of the experiment. The last mutation in EVO5 is an identical frameshift mutation in the two lineages that inactivates Rad9. Interestingly, loss of function mutations in *RAD9*, despite the large target size of this gene, only appear relatively late during the experiment (*Figure 5A* and *Figure 5—figure supplement 2*). Furthermore, they happen after other mutations have reduced some of the problems imposed by replication stress. This order suggests that a sustainable fitness advantage of mutations of the DNA damage checkpoint may depend on previous changes in the replication forks stability.

## Implications for natural evolution

Despite being conserved across much of evolution, some of the modules that collectively perform chromosome metabolism and maintain genomes show major important differences between clades, even within the eukaryotic kingdom (*Akiyoshi and Gull, 2014*; *Gourguechon et al., 2013*; *Liu et al., 2009*). For instance, a recent study found species in the yeast genus *Hanseniaspora* that lack several important genes implicated in cell cycle progress and DNA repair, including checkpoint factors such as *RAD9* and *MAD2* (*Steenwyk et al., 2019*). Trying to explain these differences is puzzling, especially if *ad-hoc* selectionist hypotheses are invoked for each different feature. For instance, what could select for a lack of an important safeguard such as the DNA damage checkpoint? Interestingly, the same lineage of *Hanseniaspora* also lacks *CDC13* (*Steenwyk et al., 2019*), an essential gene in *S. cerevisiae*, implicated in telomere replication. Studies have shown how the lethality of *cdc13Δ* mutants, is suppressed by simultaneous mutations in checkpoint factors, including *RAD9* (*Ngo and Lydall, 2010*). The evolutionary plasticity of chromosome metabolism that we reveal in this work may help to explain differences like these: mutations in ancestral cells, such as the loss of *CDC13*, could initiate an evolutionary trajectory that progressively modifies modules that are functionally linked and ultimately leads to increased fitness.

But what are the initial perturbations that trigger such changes in fundamental aspects of cell biology? The *ctf4Δ* cells that we evolved have a 25% fitness difference relative to their wild type ancestors, meaning that they would rapidly be eliminated from any population of reasonable size. Given the evolutionary rarity of major rearrangements in cell biology we can invoke events that are improbable including passing through very small populations bottlenecks or being attacked by selfish genetic elements whose molecular biology targets an important protein in an essential process. If the processes that were damaged during these events, were part of chromosome metabolism, the consequent evolutionary adaptation could lead to changes in the rates at which the structures of genomes evolve. An increase in these rates, in turn, could potentially accelerating speciation by making it easier for populations to acquire meiotically incompatible chromosome configurations.

## Implications for cancer evolution

Remarkably, our experiment recapitulates several phenomena observed during cancer development. Replication stress is thought to be a ubiquitous feature of cancer cells (*Macheret and Halazonetis, 2015*) with oncogene activation leading to replication stress and genetic instability (*Bartkova et al., 2006*; *Di Micco et al., 2006*; *Neelsen et al., 2013*). The absence of Ctf4 in our ancestor cells causes several phenotypes observed in oncogene-induced DNA replication stress including late-replicating regions, elevated mutation rates, and chromosome instability (*Fumasoni et al., 2015*; *Macheret and Halazonetis, 2015*; *Muñoz and Méndez, 2017*). Furthermore, simply by propagating cells, we generated evolved lines that mimic many features seen in tumors: (a) individual final populations contain genetically heterogeneous clones, often with different karyotypes characterized by aneuploidies and chromosomal rearrangements (*Davoli et al., 2013*; *Laughney et al., 2015*; *Lengauer et al., 1998*), (b) evolved lineages display altered DNA replication profiles compared both to WT cells and their mutant ancestors (*Amiel et al., 1999*; *Donley and Thayer, 2013*), (c) several lines have inactivated the DNA damage checkpoint (*Hollstein et al., 1991*; *Schultz et al., 2000*), and d) improved sister chromatid cohesion (*Rhodes et al., 2011*; *Sarogni et al., 2019*; *Xu et al., 2011*). All these features are adaptive in our populations, suggesting that similar changes in cancer cells may be the result of selection and contribute to the accumulation of other cancer hallmarks during cancer evolution. The similarities between tumorigenesis and our experiment lead us to speculate that a major selective force in the early stages of tumor evolution is the need to counteract the fitness costs of replication stress. Understanding the evolutionary mechanisms and dynamics of the adaptation to replication stress could therefore shed light on the early stage of tumor development.

## Perspective

In this work, we identified the main adaptive strategies that cells use to adapt to DNA replication stress induced by the absence of Ctf4. Our results reveal that defects in one function can be compensated for by two types of mutations: those in the original function and those in functions that are biologically coupled to it. Focusing on less common adaptive strategies, apparently unlinked to chromosome metabolism, could therefore potentially identify novel players that affect genome stability. It would also be interesting to induce DNA replication stress by other means, such as de-regulating replication initiation or by inducing re-replication. Analyzing the response to these challenges will reveal whether the DNA replication module has a common or diverse set of evolutionary strategies to different perturbations. Finally, this approach could be extended to many other types of cellular stress, potentially revealing other molecular adaption aspects that could collectively help understanding cellular evolution.

# Materials and methods

**Key resources table**

| Reagent type (species) or resource | Designation | Source or reference | Identifiers | Additional information |
|---|---|---|---|---|
| Strain, strain background (*S. cerevisiae*) | W303 | Murray lab | W1588 | The complete list of derived strains is available in *Supplementary file 5* |
| Commercial assay or kit | Nextera DNA Library Prep (24 Samples) | Illumina | 15028212 | |
| Commercial assay or kit | Nextera Index kit (96 Indices) | Illumina | 15028216 | |
| Commercial assay or kit | High Sensitivity D1000 ScreenTape | Agilent | 5067–5584 | |
| Software, algorithm | Python | www.python.org | RRID:SCR_008394 | Custom pipelines available at github.com/marcofumasoni/Fumasoni_and_Murray_2019 |
| Software, algorithm | Fiji | fiji.sc | RRID:SCR_002285 | |

## Strains

All strains were derivatives of a modified version (Rad5$^+$) of *S. cerevisiae* strain W303 (*leu2-3,112 trp1-1 can1-100 ura3-1 ade2-1 his3-11,15, RAD5$^+$*). *Supplementary file 5* lists each strain's genotype. The ancestors of WT and *ctf4Δ* strains were obtained by sporulating a *CTF4/ctf4Δ* heterozygous diploid. This was done to minimize the selection acting on the ancestor strains before the beginning of the experiment. Diploid stains were grown on YPD, transferred to sporulation plates (sodium acetate 0.82%, potassium chloride 0.19%, sodium chloride 0.12%, magnesium sulfate 0.035%) and incubated for four days at 25°C. Tetrads were re-suspended in water containing zymolyase (Zymo research, RRID:SCR_008968, Irvine, CA, US, 0.025 u/μl), incubated at 37°C for 45 s, and dissected on a YPD plate using a Nikon eclipse E400 microscope equipped with a TDM micromanipulator. Spores were allowed to grow into visible colonies and genotyped by presence of genetic markers and PCR.

## Media and growth conditions

Standard rich medium, YPD (1% Yeast-Extract, 2% Peptone, 2% D-Glucose) was used for all experiments except in the experiment in *Figure 4—figure supplement 4B* where YP + 2% raffinose and YP + 2% raffinose + 2% galactose were also used. Cells were synchronized either in metaphase, for 3 hr in YPD containing nocodazole (8 μg/ml, in 1% DMSO) or in G1, for 2 hr in YPD, pH 3.5 containing α-factor (3 μg/ml). Synchronization was verified by looking at cell morphology. In the experiment in *Figure 2E*, cells were then washed twice in YPD containing 50 μg/ml pronase (Zymo research) and released in S-phase at 30°C in YPD. α-factor (3 μg/ml) was added again at 30 min to prevent a second cell cycle from occurring.

## Experimental evolution

The 16 populations used for the evolution experiment were inoculated in glass tubes containing 10 ml of YPD from eight *ctf4Δ* colonies (EVO1-8) and 8 WT colonies (EVO9-16). All the colonies were derived by streaking out *MATa* (EVO1-4 and EVO25-28) or *MATα* (EVO5-9 and EVO29-32) ancestors. Glass tubes were placed in roller drums at 30°C and grown for 24 hr. Daily passages were done by diluting 10 μl of the previous culture into 10 ml of fresh YPD (1:1000 dilution, allowing for approximately 10 generations/cycle). All populations were passaged for a total of 100 cycles (≈1000 generations). Every five cycles (≈50 generations) 800 μl of each evolving population was mixed with 800 μl of 30% v/v glycerol and stored at −80°C for future analysis (*Figure 1B*). After 1000 generations four evolved clones were isolated from the each of the eight *ctf4Δ* evolved populations (a total of 32 clones) by streaking cells on a YPD plate. Single colonies were then grown in YPD media and saved in glycerol at −80°C as for the rest of the samples.

## Whole genome sequencing

Genomic DNA library preparation was performed as in *Koschwanez et al. (2013)* with an Illumina (RRID:SCR_010233, San Diego, CA, US) Nextera DNA Library Prep Kit. Libraries were then pooled and sequenced either with an Illumina HiSeq 2500 (125bp paired end reads) or an Illumina NovaSeq (150 bp paired end reads). The SAMtools software package (RRID:SCR_002105, samtools.sourceforge.net) was then used to sort and index the mapped reads into a BAM file. GATK (RRID:SCR_001876, www.broadinstitute.org/gatk; *McKenna et al., 2010*) was used to realign local indels, and VarScan (RRID:SCR_006849, varscan.sourceforge.net) was used to call variants. Mutations were found using a custom pipeline written in Python (RRID:SCR_008394, www.python.org). The pipeline (github.com/koschwanez/mutantanalysis) compares variants between the reference strain, the ancestor strain, and the evolved strains. A variant that occurs between the ancestor and an evolved strain is labeled as a mutation if it either (1) causes a non-synonymous substitution in a coding sequence or (2) occurs in a regulatory region, defined as the 500 bp upstream and downstream of the coding sequence (*Supplementary file 1*).

## Identification of putative adaptive mutations

Three complementary approaches were combined to identify the putative modules and genes targeted by selection.

## Convergent evolution on genes

This method relies on the assumption that those genes that have been mutated significantly more than expected by chance alone, represent cases of convergent evolution among independent lines. The mutations affecting those genes are therefore considered putatively adaptive. The same procedure was used independently on the mutations found in WT and ctf4Δ evolved lines:

We first calculated per-base mutation rates as the total number of mutations in coding regions occurring in a given background (ctf4Δ evolved or WT evolved), divided by the size of the coding yeast genome in bp (including 1000 bp per ORF to account for regulatory regions)

$$\lambda = \frac{SNPs + indels}{bp\,\text{coding}}$$

If the mutations were distributed randomly in the genome at a rate λ, the probability of finding n mutations in a given gene of length N is given by the Poisson distribution:

$$P\left(n\;mutations | gene\;of\;length\;N\right) = \frac{(\lambda N)^n e^{-\lambda N}}{n!}$$

For each gene of length N, we then calculated the probability of finding ≥n mutations if these were occurring randomly.

$$P(\geq n\;mutations | gene\;of\;length\;N) = \sum_{k=n}^{\infty} \frac{(\lambda N)^k e^{-\lambda N}}{k!} = 1 - \frac{\Gamma\left(n+1, \lambda N\right)}{n!}$$

(Where Γ is the upper incomplete gamma function) which gives us the p-value for the comparison of the observed mutations with the null, Poisson model. In order to decrease the number of false positives, we then performed multiple-comparison corrections. The more stringent Bonferroni correction (α=0.05) was applied on the WT evolved mutations dataset, while Benjamini-Hochberg correction (α=0.05) was used for the ctf4Δ mutation dataset. Genes that were found significantly selected in the evolved WT clones (after Bonferroni correction) were removed from the list of evolved ctf4Δ strains. This is because, since they were target of selection even in WT cells, they are likely involved in processes that are un-related to DNA replication and are instead associated with adaptation to sustained growth by serial dilutions. *Supplementary file 2* lists the mutations detected in evolved ctf4Δ clones, after filtering out those that occurred in genes that were significantly mutated in the WT populations. Genes significantly selected in these clones are shown in dark gray (after Benjamini-Hochberg correction with α=0.05). The custom pipeline used for the data analysis is available on GitHub: https://github.com/marcofumasoni/Fumasoni_and_Murray_2019 (copy archived at https://github.com/elifesciences-publications/Fumasoni_and_Murray_2019).

## Bulk segregant analysis

Bulk segregants analysis experimentally identifies putative adaptive mutations present in a given evolved clone. Briefly, a clone is selected from the population and then backcrossed to a derivative of the WT ancestor. The resulting diploid is sporulated, allowing the mutant alleles accumulated during 1000 generations to randomly segregate among the haploid progeny. The haploid progeny is then selected for growth (and for ctf4Δ) for 50–80 generations in rich media. This regime, as in the experimental evolution, selects for cells with higher fitness. The cells with causal alleles therefore quickly increase their frequency within the selected population. Non-causal alleles segregate randomly and, since they don't contribute to fitness, they are expected to be present in half of the cells at the end of the progeny selection. Deep sequencing of the genomic DNA extracted from the selected progeny population reveal the alleles frequencies and allows the identification of the ones that segregate with the evolved phenotype (frequency >70% in our case). Bulk segregant analysis was adapted from *Koschwanez et al. (2013)*. One clone per population was selected for further analysis (*Figure 1—figure supplement 2*). In these clones, the original ctf4Δ genetic marker *ble* was substituted with a *KanMX6* cassette by homologous recombination, to allow for a more efficient selection. ura3-1 evolved clones were mated with either a MATa or MATα, ura3::NatMX4-pSTE2-URA3 derivative of the WT ancestor. In this strain, the endogenous URA3 promoter is replaced with the STE2 promoter, which is only induced in MATa cells, making it possible to select for MATa

spores after meiosis. Mating was performed by mixing cells from the two strains together on a YPD plate with a toothpick and growing overnight at 30°C. The mating mixtures were then plated on double selective media, and a diploid strain from each cross was selected from a colony on the plate. To sporulate the diploid strains, cultures were grown to saturation in YPD, and then diluted 1:100 into YP 2% acetate. The cells were grown in acetate for 12 hr., pelleted and resuspended in 2% acetate. After 5 days of incubation on a roller drum at 25°C, sporulation was verified by observing the formation of tetrads under the microscope. To digest ascii, 10 ml of the sporulated culture was pelleted and resuspended in 500 µl with 250 units of Zymolyase for 1 hr. at 30°C. 4000 µl of water and 500 µl of 10% Triton X-100 were added, and the digested spores were then sonicated for 1 min to separate the tetrads. The spores were spun down slowly (6000 rpm) and resuspended in 50 ml of -URA +G418 medium. This medium selects for *ctf4Δ* haploid *MATa* cells: neither haploid *MATα* nor diploid *MATa/MATα* cells can express *URA3* from the *STE2* promoter. Each culture was then diluted 1:100 in fresh -URA + G418 medium for 10 consecutive passages, allowing for ≈66 generations to occur. Genomic DNA was extracted from the final saturated culture and used for library preparation and whole genome sequencing as described.

## Convergent evolution on modules

Statistical methods to find frequently mutated genes are focused on individual genes that contribute to an evolved trait. Functions that can be modified by affecting several genes would be therefore under-represented in the previous analysis. To account for this, we looked for gene ontology (GO) terms enriched among the mutations found to be positively selected in *ctf4Δ* evolved clones (*Supplementary file 1*, dark rows), or found segregating with the evolved phenotype by bulk segregant analysis (*Figure 1—figure supplement 2*). The combined list of mutations was input as 'multiple proteins' in the STRING database, which reports on the network of interactions between the input genes (https://string-db.org). Several GO terms describing pathways involved in the DNA and chromosome metabolism were found enriched among the putative adaptive mutations provided (*Figure 1—figure supplement 3* and *Supplementary file 3*). Since GO terms are often loosely defined and partially overlapping, we manually identified, based on literature search, four modules as putative targets of selection: DNA replication, chromosome segregation, cell cycle checkpoints, and chromatin modifiers. The full list of mutated genes observed in the evolved *ctf4Δ* clones was then used as input in the STRING database (RRID:SCR_005223). This was done to account for genes, that despite not being identified as containing adaptive mutations by the previous techniques, are part of modules under selection: mutations in these genes could have contributed to the final phenotype. The interaction network between mutated genes was downloaded and curated in Cytoscape (RRID:SCR_003032, https://cytoscape.org/). For clarity of representation, only those nodes strongly connected to the previously identified modules are shown in *Figure 1D*.

## Fitness assays

To measure relative fitness, we competed the ancestors and evolved strains against reference strains. Both WT (*Figure 1C*, *Figure 1—figure supplement 1*, *Figure 4—figure supplement 4A*, *Figure 5—figure supplement 2*) and *ctf4Δ* (*Figure 2D*, *Figure 2—figure supplement 1C*, *Figure 3C*, *Figure 4C*, *Figure 4—figure supplement 4A–B*, *Figure 5A*) reference strains were used. A *pFA6a-prACT1-yCerulean-HphMX4* plasmid was digested with *AgeI* and integrated at one of the *ACT1* loci of the original heterozygous diploid (*CTF4/ctf4Δ*) strain. This allow for the expression of fluorescent protein yCerulean under the strong actin promoter. The heterozygous diploid was then sporulated and dissected to obtain fluorescent WT or *ctf4Δ* reference haploid strains. For measuring the relative fitness, 10 ml of YPD were inoculated in individual glass tubes with either the frozen reference or test strains. After 24 hr. the strains were mixed in fresh 10 ml YPD tubes at a ratio dependent on the expected fitness of the test strain compared to the reference (i.e. 1:1 if believed to be nearly equally fit) and allowed to proliferate at 30°C for 24 hr. 10 µl of samples were taken from this mixed culture (day 0) and the ratio of the two starting strains was immediately measured. Tubes were then cultured following in the same conditions as the evolution experiment by diluting them 1:1000 into fresh medium every 24 hr for 4 days, monitoring the strain ratio at every passage. Strain ratios and number of generations occurred between samples were measured by flow cytometer

(Fortessa, BD Bioscience, RRID:SCR_013311, Franklin Lakes, NJ, US). Ratios $r$ were calculated based on the number of fluorescent and non-fluorescent events detected by the flow cytometer:

$$r = \frac{NonFluorescent_{events}}{Fluorescent_{events}}$$

Generations between time points $g$ were calculated based on total events measured at time 0 hr. and time 24 hr.:

$$g = \frac{\log_{10}(events_{t24}/events_{t0})}{\log_{10} 2}$$

Linear regression was performed between the $(g, \log_e r)$ points relative to every sample. Relative fitness was calculated as the slope of the resulting line. The mean relative fitness $s$ was calculated from measurements obtained from at least three independent biological replicates. Error bars represent standard deviations. The P-values reported in figures are the result of $t$-tests assuming unequal variances (Welch's test). Note that the absolute values of relative fitness change depending on the reference strain used: a strain that shows 27% increased fitness when measured against $ctf4\Delta$ (that is 27% less fit then WT), does not equate the WT fitness. This is because a 27% increase of 0.73 ($ctf4\Delta$ fitness compared to WT) gives 0.93, hence a 7% fitness defect compared to WT.

## Cell cycle profiles

Cell cycle analysis was conducted as previously described (Fumasoni et al., 2015). In brief, $1 \times 10^7$ cells were collected from cultures by centrifugation, and resuspended in 70% ethanol for 1 hr. Cells were then washed in 50 mM Tris-HCl (pH 7.5), resuspended in the same buffer containing 0.4 μg/ml of RNaseA and incubated at 37°C for at least 2 hr. Cells were collected and further treated overnight at 37°C in 50 mM Tris-HCl (pH 7.5) containing proteinase K (0.4 μg/ml). Cells were then centrifuged and washed in 50 mM Tris-HCl (pH 7.5). Samples were then diluted 10–20-fold in 50 mM Tris-HCl (pH 7.8) containing 1 mM SYTOX green, and analyzed by flow cytometer (Fortessa, BD Bioscience). The FITC channel was used to quantify the amounts of stained-DNA per cell. 10000 events were acquired for each sample. Cell cycle profiles were analyzed and visualized in FlowJo (RRID:SCR_008520, BD Bioscience). The percentage of genome replicated at 30 min was calculated based on the cell cycle profile as follow $G_{rep} = DNA\ content\ mode/(2C - 1C) * 100$. The height of the 1C and 2C peaks was obtained as the max cells count reached by the respective peak. For both the percentage of genome replicated at 30 min and the 1C/2C ratio, the mean was calculated from values obtained with three independent biological replicates. Error bars represent standard deviations. The P-values reported are the result of $t$-tests assuming unequal variances.

## Copy number variations (CNVs) detection by sequencing

Whole genome sequencing and read mapping was done as previously described. The read-depths for every unique 100 bp region in the genome were then obtained by using the VarScan copynumber tool. A custom pipeline written in python was used to visualize the genome-wide CNVs. First, the read-depths of individual 100 bp windows were normalized to the genome-wide median read-depth to control for differences in sequencing depths between samples. The coverage of the ancestor strains was then subtracted from the one of the evolved lines to reduce the noise in read depth visualization due to the repeated sequences across the genome. The resulting CNVs were smoothed across five 100 bp windows for a simpler visualization. Final CNVs were then plotted relative to their genomic coordinate at the center of the smoothed window. Since the WT CNVs were subtracted from the evolved CNVs, the y axis refers to the copy number change occurred during evolution (i.e. +1 means that one an extra copy of a chromosome fragment has been gained). The custom pipeline used for the data analysis is available on GitHub: https://github.com/marcofumasoni/Fumasoni_and_Murray_2019.

## Premature sister chromatid separation assay

Logarithmically growing cells were arrested in metaphase as previously described. Samples were then collected and fixed in 4% formaldehyde for 5 min at room temperature. Cells were washed In SK buffer (1M sorbitol, 0.05 M $K_2PO_4$) and sonicated for 8 s prior to microscope analysis. Images

were acquired with a Nikon eclipse Ti spinning-disk confocal microscope using a 100X oil immersion lens. Fluorescence was visualized with a conventional FITC excitation filter and a long pass emission filter. Images were analyzed using Fiji (RRID:SCR_002285, https://fiji.sc/). At least 100 cells were analyzed to calculate the percentage of premature chromatid Separation for each strain. The mean value was calculated from measurements obtained with three independent biological replicates. Error bars represent standard deviations. The P-values reported are the result of t-tests assuming unequal variances.

## DNA replication profiles

DNA replication profiling was adapted from *Müller et al. (2014)*; *Saayman et al., 2018*; *Bar-Ziv et al. (2016)*. Genomic DNA and library preparation were performed independently on all the collected samples as previously described. Repeated sequences (such as telomeres, rDNA and Ty elements) were excluded from the CNV analysis as non-uniquely mapped reads can alter local read-depth and generate artefacts. A custom python script was used to analyze the CNVs from multiple time points from the same strain to produce DNA replication profiles. Read-depths of individual 100 bp windows were normalized to the genome-wide median read-depth to control for differences in sequencing depths between consecutive samples. To allow for intra-strain comparison, coverage was then scaled according to the sample DNA content measured as the median of the cell-cycle profile obtained by flow cytometry. The resulting coverage was then averaged across multiple 100 bp windows and a polynomial data smoothing filter (Savitsky-Golay) was applied to the individual coverage profiles to filter out noise. Replication timing $t_{rep}$ is defined as the time at which 50% of the cells in the population replicated a given region of the genome (*Figure 4—figure supplement 2*), which is equivalent to an overall relative coverage of 1.5x, since 1x corresponds to an unreplicated region and 2x to a fully replicated one. The replication timing $t_{rep}$ was calculated by linearly interpolating the two time points with coverage lower and higher than 1.5x and using such interpolation to compute the time corresponding to 1.5x coverage. Final $t_{rep}$ were then plotted relative to their window genomic coordinates. Unreplicated regions at 45 min were calculated as the sum of all regions with $t_{rep}$ >45 min. To find DNA replication origins, the $t_{rep}$ profiles along the genome were filtered using a Fourier low-pass filter to remove local minima and then used to find local peaks. Only origins giving rise to long replicons were used to measure fork velocity. Fork velocity was calculated by dividing the distance between the origin and the closest termination site by the time required to replicate the region. Duplicate replication profiles were obtained from two experiments performed on biological replicates. Reproducibility was confirmed with qualitatively and quantitatively comparable results across duplicates. The data obtained from the first duplicate are reported. The reliability of the pipeline was assessed by qualitatively and quantitatively comparing our WT results with previously reported measurements (*Müller et al., 2014*; *Raghuraman et al., 2001*). The custom pipeline used for the data analysis is available on GitHub: https://github.com/marcofumasoni/Fumasoni_and_Murray_2019.

## The correlation of chromosomal features with fork-stall zones

We first identified chromosomal locations where fork stalling in the ancestral ctf4Δ cells prevented the completion of DNA replication by 45 min (fork-stall zones). The fork position at 45 min was considered the center of the fork-stall zones, while 5 kb upstream and downstream the fork site were included in the analysis to account for features in the proximity of the fork that could have interfered with its progression. We then examined the sequences within these windows to determine whether various chromosomal features were over- or underrepresented. We considered features that previous studies have found to be associated with hotspots for lesions and sources of genetic instability. We first counted how many times a given feature fell in a fork-stall zone. Then we calculated the expected number of features in these zones based on the total number of features in the genome and the percentage of the genome represented by fork-stall zones. We compared these numbers by χ2 analysis and reported the associated p-values (*Supplementary file 4*). The number of tRNA genes, transposable elements, LTRs, ARS elements, snRNA and snoRNA genes and centromeres in the genome were determined using YeastMine (https://yeastmine.yeastgenome.org/). G4 sequences were obtained from *Capra et al. (2010)*. Highly-(top 5%) and weakly-(least 5%) transcribed genes were identified from the data in *Nagalakshmi et al. (2008)*. Rrm3 binding sites and regions with

high levels of γH2AX were derived from *Azvolinsky et al. (2006)* and *Szilard et al., 2010*, respectively. Site of DNA replication termination were derived from valleys in the $t_{Rep}$ signal of the wild type strain (*Figure 4—figure supplement 3*, green signal). The tandemly repeated sequences, with a minimal repeat tract of twenty-four bases, were obtained from the tandem-repeat-database (TRDB; https://tandem.bu.edu/cgi-bin/trdb/trdb.exe).

## Analysis of allele frequency by sanger sequencing

Allele frequencies within populations were estimated as in *Wildenberg and Murray (2014)*. In brief, chromatograms obtained by sanger sequencing were used to estimate the fraction of mutant alleles in a population at different time points during the evolution. The fraction of mutant alleles in the population was assumed to be the height of the mutant allele peak divided by the height of the mutant allele peak plus the ancestor allele peak. The values from two independent sanger sequencing reactions, obtained by primers lying upstream and downstream the mutations, were averaged to obtain the final ratios. Error bar edges represent the ratios obtained by the two independent sequencing reactions. Values below the approximate background level were assumed to be zero, and values above 95% were assumed to be 100%.

## Segmental amplification detection by digital PCR

Droplet digital PCR was used to detect the amplifications of the fragment containing *SCC2* at different time points during evolution. Genomic DNA was prepared and diluted accordingly. Bio-Rad (RRID:SCR_008426, Hercules, CA, US) ddPCR supermix for probes (no dUTP) was used to prepare probes specific to *SCC2* and the centromere of chromosome IV. A Bio-Rad QX200 Droplet Generator was used to generate droplets containing genomic DNA and probes. The droplet PCR was performed in a Bio-Rad thermocycler and analyzed with a Bio-Rad QX200 Droplet Reader. At least 10,000 droplets were acquired for each strain. Analysis was performed on biological duplicates with comparable results. Data obtained with the first duplicate are shown. Droplet analysis was performed with QuantaSoft software (Bio-Rad). SCC2/Centromere ratios were then used to quantify SCC2 copy numbers. Error bars represent Poisson 95% confidence intervals. To estimate the percent of cells carrying the SCC2 amplification within a population we assumed that the allele spreading in the population was a duplication of SCC2 (as indicated by the EVO5 copy number analysis). Values above 95% were assumed to be 100%.

## Acknowledgements

We thank Stephen Elledge and Philip Zegerman for sharing yeast strains; Andrea Giometto, Mayra Garcia and John Koschwanez for assistance in data analysis; Stephen Bell, Michael Desai, Michael Laub, Bodo Stern, Angelika Amon, Sriram Srikant, Thomas LaBar, Miguel Coelho and Yi Chen for critical reading of the manuscript; Claire Hartman and Zachary Niziolek from the Harvard Bauer Core Facility for technical assistance. Yoav Voichek and Felix Jonas for advice on DNA replication profiling; We thank the members of the Murray and Nelson labs for helpful discussions. MF gratefully acknowledges fellowship support from the Human Frontiers Science Program, EMBO and AIRC.

## Additional information

### Funding

| Funder | Grant reference number | Author |
|--------|------------------------|--------|
| Human Frontier Science Program | LT000786/2016-L | Marco Fumasoni |
| European Molecular Biology Organization | ALTF 485-2015 | Marco Fumasoni |
| Associazione Italiana per la Ricerca sul Cancro | iCARE 17957 | Marco Fumasoni |
| National Institute of General Medical Sciences | RO1-GM43987 | Andrew W Murray |

| National Science Foundation | #1764269 | Andrew W Murray |
| Simons Foundation | #594596 | Andrew W Murray |

The funders had no role in study design, data collection and interpretation, or the decision to submit the work for publication.

## Author contributions

Marco Fumasoni, Conceptualization, Data curation, Software, Formal analysis, Funding acquisition, Investigation, Writing - original draft, Project administration, Writing - review and editing; Andrew W Murray, Conceptualization, Supervision, Funding acquisition, Writing - original draft, Project administration, Writing - review and editing

## Author ORCIDs

Marco Fumasoni https://orcid.org/0000-0002-4507-7824
Andrew W Murray http://orcid.org/0000-0002-0868-6604

## Decision letter and Author response

Decision letter https://doi.org/10.7554/eLife.51963.sa1
Author response https://doi.org/10.7554/eLife.51963.sa2

# Additional files

## Supplementary files

• Supplementary file 1. Mutations detected in the evolved lines.

• Supplementary file 2. Putative adaptive mutations in evolved *ctf4Δ* strains.

• Supplementary file 3. Enriched GO terms among putative genes under positive selection in evolved *ctf4Δ* strains.

• Supplementary file 4. Chromosome features enriched in fork-stall zones in *ctf4Δ* cells.

• Supplementary file 5. Yeast strains used in the study.

• Supplementary file 6. Tables represented in figures.

• Transparent reporting form

## Data availability

A major dataset, containing the sequencing data used in the manuscript has been made publicly available at the EBI European Nucleotide Archive (Accession no: PRJEB34641).

The following dataset was generated:

| Author(s) | Year | Dataset title | Dataset URL | Database and Identifier |
|---|---|---|---|---|
| Fumasoni M, Murray AW | 2020 | The evolutionary plasticity of chromosome metabolism allows adaptation to DNA replication stress | https://www.ebi.ac.uk/ena/browser/view/PRJEB34641 | EBI European Nucleotide Archive, PRJEB34641 |

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
