## [Decision Letter]

**Acceptance summary:**

Your paper is a beautiful example of how the fitness effects due to perturbation of a conserved gene can be surprisingly easily, at least in lab conditions, mitigated by compensatory mutations, and how these suppressor mutations are unpredictable and in turn provide new insight into the function of the originally perturbed gene.

**Decision letter after peer review:**

Thank you for submitting your article "The evolutionary plasticity of chromosome metabolism allows adaptation to DNA replication stress" for consideration by *eLife*.

Your article has now been reviewed by three peer reviewers, one of whom is a member of our Board of Reviewing Editors, and the evaluation has been overseen by Jessica Tyler as the Senior Editor. The following individuals involved in review of your submission have agreed to reveal their identity: Anna Selmecki (Reviewer #2); Conrad Nieduszynski (Reviewer #3).

The reviewers have discussed the reviews with one another and the Reviewing Editor has drafted this decision to help you prepare a revised submission.

All reviewers agree that the paper is interesting, constitutes a large body of work and is exceptionally well-written. On the other hand, the reviewers also struggled a bit to answer the question why this study belongs in *eLife*. After discussion of this, it became clear that part of the problem might be in the fact that the paper falls a bit between a study on evolutionary mechanisms and a study on replication stress. This is of course not a problem in itself and could in fact be a strong point. However, in this case, it proved more difficult to grasp the central question, the main conclusions, and their novelty.

Encouragingly, the cover letter separates both aspects quite well, and seems to highlight the evolutionary angle as the key message. It is known that (isolated) lineages can overcome severe fitness defects related to gene deletion; including deletion of highly conserved genes that are marked as "essential". However, as you point out in the cover letter, it is true that your study does a remarkable job in revealing the actual suppressor mutations and their interactions, but to the reviewers, this message is lost a bit in the paper, possibly because the Introduction starts more from the angle of chromosome biology.

On the other hand, the reviewers felt that results are perhaps less conclusive when it comes to yielding insight into the process of replication. While the results show how different compensatory mutations in cohesion loading, cell cycle checkpoints and replication for progression can mitigate the loss of Ctf4, the reviewers agree that these aspects have perhaps not been studied in sufficient detail to be sure about the underlying mechanism and the role of Ctf4, and that they could therefore perhaps be more critically discussed and grouped in a separate paragraph in the Discussion section.

In addition, we would ask you to take the other main comments of the reviewers into account. Specifically, both reviewer 1 and 2 ask to better highlight previous studies that have examined the evolutionary routes that compensate gene loss, and explain better how this paper goes beyond these previous studies.

Reviewer #1:

This paper investigates the evolutionary routes that mitigate the effect of inactivation of Ctf4, a protein that coordinates replication fork activity in the model yeast *Saccharomyces cerevisiae*. The results show that after 1000 generations, different lines find distinct mutational paths to compensate for the loss of Ctf4, but while the exact driver mutations differ, each path combines changes in DNA replication, DNA damage checkpoint and chromatid cohesion.

This is a solid and beautifully written study that provides new insight into evolutionary response to mutations in crucial and highly conserved parts of the DNA replication machinery. Specifically, it is interesting to see how parallel evolution hints at the interplay between different aspects of DNA replication (replication fork progression, DNA damage checkpoints, chromosome alignment and cohesion). Perhaps equally importantly, the authors also show some similarities to the mutational trajectories of some tumors in higher eukaryotes.

The most important question is what truly novel insight this study brings. The conclusions touch upon different fields (evolution, DNA replication and cancer), but it is unclear if the results really provide sufficient novel and general insight into any of these to merit publication in *eLife*. Several studies have evolved lines in which an important (or even "essential") gene was inactivated, and it has already been shown that cells can often evolve suppressor mutations. Moreover, the mutational paths found in this study seem highly unlikely in natural populations because the Ctf4 deletion strain suffers such a high fitness defect. While there are some striking parallels with tumor progression, it is also unclear whether the deletion of Ctf4 really provides a solid model for replication stress and the mutational path in tumors. Lastly, whereas the compensatory mutations provide some clues into the function of Ctf4 (productive loading of cohesin?), these would need to be verified.

So, in a way, while the paper touches upon several very interesting basic questions and important phenomena, none of these are followed through in detail. I therefore wonder whether it would be better to pick one of the 3 research lines and really focus on that one to dig a bit deeper, with some obvious additional follow-up experiments and deeper discussion? Obviously, this does not imply that there would not be room to also point out parallels with the other phenomena in the Discussion section; but by picking one major line, perhaps the paper would feel more like a complete story and would in fact draw more instead of less attention?

Reviewer #2:

The mechanistic machinery that performs DNA replication is highly conserved throughout organismal evolution, yet we still lack fundamental knowledge of key machinery as well as an overall understanding for how these systems evolved and are maintained. While it's not surprising that adaptation occurred so rapidly for the ancestral ctf4 strain, because of its massive fitness defect, it is fascinating how the evolved lineages sequentially modified chromosome cohesion, altered the speed of replication forks, and lost a key cell-cycle regulator of DNA damage. Experimental evolution enables counter-intuitive or detrimental solutions to be explored, while only successful mutant combinations in the correct order are selected. The manuscript was well written and easily understandable, often with definitions inserted right where they were helpful.

Reviewer #3:

The authors have undertaken an experimental evolution study in budding yeast cells experiencing constitutive DNA replication stress. The stress was a consequence of the deletion of a non-essential replication factor, CTF4. Interestingly the authors find that within 1000 generation (and three adaptive steps) cells have recovered much of the fitness cost of CTF4-deletion. The three steps are segmental duplication of the cohesion loader SCC2, mutations associated with reduced replication fork velocity and inactivation RAD9 (a checkpoint mediator).

Overall, the findings are generally convincing with potentially important parallels to speciation and cancer development. In my opinion the greatest weakness of this study is the degree to which the authors have had to infer how the discovered mutations might mechanistically give adaptive advantage. For example, increased copy number of SCC2 is implied to lead to greater cohesin loading, however the authors haven't directly shown greater cohesin loading (although to their credit they have shown reduced premature chromatid separation). The SLD5 and IXR1 mutations are proposed to give reduced fork velocity, but this isn't demonstrated. Loss of RAD9 is suggested to allow post-mitotic completion of DNA replication, but again this isn't demonstrated (e.g. RPA foci number increased). I think firming up these inferred mechanisms is probably beyond the scope of this study, but I'm still slightly disappointed that the authors weren't able to make more progress on this given how clearly they present these models in Figure 6.

[Editors' note: further revisions were suggested prior to acceptance, as described below.]

Thank you for resubmitting your work entitled "The evolutionary plasticity of chromosome metabolism allows adaptation to constitutive DNA replication stress" for further consideration by *eLife*. Your revised article has been evaluated by Jessica Tyler (Senior Editor) and a Reviewing Editor, as well as one of the original reviewers.

We appreciate your efforts in response to our concerns. The manuscript has been much improved, but we would still encourage you to more explicitly connect the hypotheses, experiments and discussion; and to stress the main conclusions and novel insight into the evolution of conserved genes (for details, see the comments of reviewer #2 below).

Reviewer #1:

I appreciate the authors' efforts to address our concerns. I believe the text now better reflects the main impact of the study and I support publication in *eLife*.

Reviewer #2:

Overall the reviewers addressed the majority of my critiques sufficiently.

However, the broader concerns of the review group, specifically the impact of the observations about the 'overall process of evolutionary adaptation' were addressed in a less-compelling way. The authors state, "Our goal, however, was to provide a comprehensive account of mutations interact with each other to explain the overall process of evolutionary adaptation." (I assume this should say of how mutations interact, or what mutations interact?) Either way, there is conflict with this statement and the proposed hypothesis. This hypothesis is not addressed with the experiments.

The hypothesis: "One hypothesis is that because so many replication proteins are essential, the observed differences can only be obtained by extremely slow evolutionary processes that require many successive mutations of small effect and happen over millions of generations. Alternatively, the DNA replication module could accommodate substantial changes within hundreds or thousands of generations, but in order to explain the overall conservation of DNA replication, such events would have to be rare compared similar phenomena affecting less conserved modules. To distinguish between these two hypotheses, we followed the evolutionary response to a genetic perturbation of DNA replication."

The experiments: using a ctf4 mutant with extremely low fitness, do not address the differences in rates or frequency of mutations acquired during the evolutionary process. These experiments determined that evolution could occur, and mutations were acquired within 100-1000 generations, but whether these are rare or not was not determined (compared to similar phenomena perhaps affecting a less conserved module with the same fitness defect, for example). I find the point that DNA replication module can evolve seems obvious (is has occurred in some examples provided!). If it was shown to be non-evolvable in previous studies, what should readers conclude from the comparison of the current study and this previous study? What have we learned about the proposed hypotheses?

---

## [Author Response]

All reviewers agree that the paper is interesting, constitutes a large body of work and is exceptionally well-written. On the other hand, the reviewers also struggled a bit to answer the question why this study belongs in eLife. After discussion of this, it became clear that part of the problem might be in the fact that the paper falls a bit between a study on evolutionary mechanisms and a study on replication stress. This is of course not a problem in itself and could in fact be a strong point. However, in this case, it proved more difficult to grasp the central question, the main conclusions, and their novelty.Encouragingly, the cover letter separates both aspects quite well, and seems to highlight the evolutionary angle as the key message. It is known that (isolated) lineages can overcome severe fitness defects related to gene deletion; including deletion of highly conserved genes that are marked as "essential". However, as you point out in the cover letter, it is true that your study does a remarkable job in revealing the actual suppressor mutations and their interactions, but to the reviewers, this message is lost a bit in the paper, possibly because the Introduction starts more from the angle of chromosome biology.On the other hand, the reviewers felt that results are perhaps less conclusive when it comes to yielding insight into the process of replication. While the results show how different compensatory mutations in cohesion loading, cell cycle checkpoints and replication for progression can mitigate the loss of Ctf4, the reviewers agree that these aspects have perhaps not been studied in sufficient detail to be sure about the underlying mechanism and the role of Ctf4, and that they could therefore perhaps be more critically discussed and grouped in a separate paragraph in the Discussion section.In addition, we would ask you to take the other main comments of the reviewers into account. Specifically, both reviewer 1 and 2 ask to better highlight previous studies that have examined the evolutionary routes that compensate gene loss, and explain better how this paper goes beyond these previous studies.

Thank you for the careful and through reviews our article. We are very pleased to hear that the reviewers found our work solid, interesting and well written. We are especially grateful to the reviewers for pointing out that the narrative structure of the manuscript generated some confusion regarding the motivation and the key findings of our work.

We have edited the paper to try to better communicate a series of points, which we summarize below:

The main motivation of our work is to investigate a central question in evolutionary biology: How can conserved cellular processes, which are essential for life, undergo significant, mechanistic changes during evolution without destroying the overall process.

To answer this question, we focused on DNA replication for three reasons. First, because DNA replication is one of the most conserved processes in life, and investigating its evolutionary plasticity has far-reaching implications in several fields of biology. Second, because previous systematic studies that identified defects that could rapidly be repaired by adaptive evolution reported that defects in DNA replication were harder to repair than those in other essential processes. Third, by focusing on a single process, we could reach a detailed, mechanistic understanding that would complement larger-scale, but less detailed studies.

Our goal was to investigate the cellular response to a long-term perturbation in DNA replication rather than to perform a detailed study of the mechanism of DNA replication. Thus, we deleted *CTF4* not to investigate its role in DNA replication, but rather as an experimental tool to induce constitutive replication stress. This allowed us to answer the following questions: i) how rapidly and how completely does fitness increase after a severe perturbation, ii) how reproducible are the evolutionary trajectories of replicate populations, iii) what genes are mutated, and which function modules do they affect, iv) what are the molecular mechanisms of adaptation, v) how do distinct mechanisms interact with each other, and vi) what do the mechanistic changes reveal about the evolutionary plasticity of the perturbed process.

We have successfully answered all these questions. By doing so, we have provided a uniquely detailed account of the molecular mechanisms that produce adaptive evolution. We argue that coupling the genetic dissection of the mutations that increase fitness to a cell biological understanding of their mechanism is essential to fully understand evolutionary mechanisms. We agree that if we were going to investigate each individual adaptive mutation in depth, other experiments could have been performed to reach the level of a paper whose primary focus was the role of individual proteins in the process of chromosome metabolism. Our goal, however, was to provide a comprehensive account of mutations interact with each other to explain the overall process of evolutionary adaptation. We believe that dissecting the effect of each mutation, in greater detail, would have made it difficult for readers to grasp the central point of our paper.

We have now modified the manuscript in several sections with the aim of effectively communicating the scope, conclusions and novelty of our article.

Reviewer #1:This paper investigates the evolutionary routes that mitigate the effect of inactivation of Ctf4, a protein that coordinates replication fork activity in the model yeast *Saccharomyces cerevisiae*. The results show that after 1000 generations, different lines find distinct mutational paths to compensate for the loss of Ctf4, but while the exact driver mutations differ, each path combines changes in DNA replication, DNA damage checkpoint and chromatid cohesion.This is a solid and beautifully written study that provides new insight into evolutionary response to mutations in crucial and highly conserved parts of the DNA replication machinery. Specifically, it is interesting to see how parallel evolution hints at the interplay between different aspects of DNA replication (replication fork progression, DNA damage checkpoints, chromosome alignment and cohesion). Perhaps equally importantly, the authors also show some similarities to the mutational trajectories of some tumors in higher eukaryotes.The most important question is what truly novel insight this study brings. The conclusions touch upon different fields (evolution, DNA replication and cancer), but it is unclear if the results really provide sufficient novel and general insight into any of these to merit publication in eLife. Several studies have evolved lines in which an important (or even "essential") gene was inactivated, and it has already been shown that cells can often evolve suppressor mutations. Moreover, the mutational paths found in this study seem highly unlikely in natural populations because the Ctf4 deletion strain suffers such a high fitness defect. While there are some striking parallels with tumor progression, it is also unclear whether the deletion of Ctf4 really provides a solid model for replication stress and the mutational path in tumors. Lastly, whereas the compensatory mutations provide some clues into the function of Ctf4 (productive loading of cohesin?), these would need to be verified.So, in a way, while the paper touches upon several very interesting basic questions and important phenomena, none of these are followed through in detail. I therefore wonder whether it would be better to pick one of the 3 research lines and really focus on that one to dig a bit deeper, with some obvious additional follow-up experiments and deeper discussion? Obviously, this does not imply that there would not be room to also point out parallels with the other phenomena in the Discussion section; but by picking one major line, perhaps the paper would feel more like a complete story and would in fact draw more instead of less attention?

Above we discuss how we have attempted to better communicate, following reviewer 3’s advice, our goal of providing a detailed, mechanistic account of how adaptive evolution can repair substantial damage to a strongly conserved and essential cellular process.

In terms of the reviewer’s criticisms that we would have been better focusing in a single area, we believe that the paper makes original contributions in all three areas they discuss: 1) although other papers have reported on the evolutionary repair of deleterious mutations, only some of these have identified the mutations responsible for repair, and none, to our knowledge, have provided the level of mechanistic explanation of how the mutations increase fitness, 2) we agree that the large fitness defect associated with the lack of Ctf4 would be strongly selected against in nature, but there are numerous cases (such as the absence of *CDC13* in *Hanseniaspora* or the anaphase promoting complex in *Giardia*, which we now mention and cite) that would be expected to generate similar fitness defects, and 3) although we agree that it is unclear how accurate a yeast *ctf4∆* strain is as a model for replication stress in tumors, the inaccessibility of evolution within tumors means that the parallels between the changes that we saw and those that have been observed in tumors are worth pointing out.

Reviewer #2:The mechanistic machinery that performs DNA replication is highly conserved throughout organismal evolution, yet we still lack fundamental knowledge of key machinery as well as an overall understanding for how these systems evolved and are maintained. While it's not surprising that adaptation occurred so rapidly for the ancestral ctf4 strain, because of its massive fitness defect, it is fascinating how the evolved lineages sequentially modified chromosome cohesion, altered the speed of replication forks, and lost a key cell-cycle regulator of DNA damage. Experimental evolution enables counter-intuitive or detrimental solutions to be explored, while only successful mutant combinations in the correct order are selected. The manuscript was well written and easily understandable, often with definitions inserted right where they were helpful.

We thank reviewer #2 for having grasped what we initially fell short in communicating.

Reviewer #3:The authors have undertaken an experimental evolution study in budding yeast cells experiencing constitutive DNA replication stress. The stress was a consequence of the deletion of a non-essential replication factor, CTF4. Interestingly the authors find that within 1000 generation (and three adaptive steps) cells have recovered much of the fitness cost of CTF4-deletion. The three steps are segmental duplication of the cohesion loader SCC2, mutations associated with reduced replication fork velocity and inactivation RAD9 (a checkpoint mediator).Overall, the findings are generally convincing with potentially important parallels to speciation and cancer development. In my opinion the greatest weakness of this study is the degree to which the authors have had to infer how the discovered mutations might mechanistically give adaptive advantage. For example, increased copy number of SCC2 is implied to lead to greater cohesin loading, however the authors haven't directly shown greater cohesin loading (although to their credit they have shown reduced premature chromatid separation). The SLD5 and IXR1 mutations are proposed to give reduced fork velocity, but this isn't demonstrated. Loss of RAD9 is suggested to allow post-mitotic completion of DNA replication, but again this isn't demonstrated (e.g. RPA foci number increased). I think firming up these inferred mechanisms is probably beyond the scope of this study, but I'm still slightly disappointed that the authors weren't able to make more progress on this given how clearly they present these models in Figure 6.

We are glad that reviewer 3 appreciates the parallels between our work and speciation and cancer. We agree that rigorous confirmation of a variety of inferences we make would require more detailed experiments, but like the reviewer, we feel that this additional work is beyond the scope of this study. We have rewritten parts of the discussion to make it as clear as possible where we can make conclusions, where we can make reasonable, Ockham’s razor-based inferences, and where we are speculating.

[Editors' note: further revisions were suggested prior to acceptance, as described below.]

We appreciate your efforts in response to our concerns. The manuscript has been much improved, but we would still encourage you to more explicitly connect the hypotheses, experiments and discussion; and to stress the main conclusions and novel insight into the evolution of conserved genes (for details, see the comments of reviewer #2 below).Reviewer #2:Overall the reviewers addressed the majority of my critiques sufficiently.However, the broader concerns of the review group, specifically the impact of the observations about the 'overall process of evolutionary adaptation' were addressed in a less-compelling way. The authors state, "Our goal, however, was to provide a comprehensive account of mutations interact with each other to explain the overall process of evolutionary adaptation." (I assume this should say of how mutations interact, or what mutations interact?)

We apologize for the typo. We meant to write ‘how mutations interact’, as investigated in the Results section ‘Epistatic interactions among adaptive mutations dictate evolutionary trajectories’ and commented in the Discussion.

Either way, there is conflict with this statement and the proposed hypothesis. This hypothesis is not addressed with the experiments.The hypothesis: "One hypothesis is that because so many replication proteins are essential, the observed differences can only be obtained by extremely slow evolutionary processes that require many successive mutations of small effect and happen over millions of generations. Alternatively, the DNA replication module could accommodate substantial changes within hundreds or thousands of generations, but in order to explain the overall conservation of DNA replication, such events would have to be rare compared similar phenomena affecting less conserved modules. To distinguish between these two hypotheses, we followed the evolutionary response to a genetic perturbation of DNA replication."The experiments: using a ctf4 mutant with extremely low fitness, do not address the differences in rates or frequency of mutations acquired during the evolutionary process. These experiments determined that evolution could occur, and mutations were acquired within 100-1000 generations, but whether these are rare or not was not determined (compared to similar phenomena perhaps affecting a less conserved module with the same fitness defect, for example).

We agree that we did not compare the rate at which mutations allow for the recovery from genetic pathways to different modules. We now state a simpler hypothesis that our experiments did test:

“One hypothesis is that the overall organization of DNA replication can change as a consequence of accumulating several mutations, each perturbing a single aspect of replication, in response to a severe initial perturbation.”

I find the point that DNA replication module can evolve seems obvious (is has occurred in some examples provided!).

We agree with this observation, as previously included in the manuscript:

“These differences reveal that although the DNA replication module performs biochemically conserved reactions, its features can change during evolution.”

As we now make clear with the revised hypothesis, our question is not whether such changes can occur but whether if they can occur rapidly in response to a severe perturbation of the module.

If it was shown to be non-evolvable in previous studies, what should readers conclude from the comparison of the current study and this previous study? What have we learned about the proposed hypotheses?

We argue that the fundamental difference between our work and previous studies was the magnitude of the fitness effects that previous studies required. For suppressor screens (Liu et al., 2015), single mutations must be capable of overcoming lethality and for the large scale, systematic studies of genetic interactions (van Leeuwen et al., 2016), the mutations must have effects larger than the noise in the fitness measurements. In contrast, the only requirement in our work is that the cumulative effect of multiple mutations is sufficient to substantially increase reproductive fitness. We have added the following paragraph to make this point.

“Previous studies have argued that perturbations in DNA replication are less likely to be repaired by single compensatory mutations than other processes, such as intracellular trafficking. We believe that the explanation for this difference between these studies and our own lies in the different nature of the mutations that are selected. Suppressor screens rely on single mutations that can either rescue lethality or whose fitness effect is greater than the noise in systematic analyses of genetic interactions. In contrast, experimental evolution following non-lethal perturbations allows for the sequential acquisitions of small-effect mutations that collectively rescue the perturbed process. Our experiment suggests that although the single mutations that can fully repair genetic damage to DNA replication are rare, the existence of combination of small-effect mutations that can repair perturbations make it evolutionarily plastic.”